**communications** engineering

# How task-relevant vibratory feedback from an active exoskeleton can lead to ergonomic postures

Waldez Gomes[1,2], Lucas Quesada[1,3], Bastien Berret [1], Nicolas Vignais [1,4] & Dorian Verdel [1,3,5] ✉

In the past decades, active exoskeletons have been dedicated to reducing human effort, in particular to assist workers in occupational environments. However, this approach does not promote the learning of more ergonomic postures by workers, which is critical for the long-term prevention of musculoskeletal disorders. Alternatively, we propose the use of exoskeletons as biofeedback systems, generating task-relevant perturbations guiding users towards ergonomic postures. To test this approach, participants performed reach-to-hold movements towards a redundant target, allowing multiple final postures. We then introduced vibrations with posture-dependent intensity, generating a sensorimotor disturbance that canceled out either above or below each participant's nominal preferred posture. Interestingly, participants adapted to minimize the vibrations, whether it increased or decreased the gravity efforts, and retained the novel posture when it induced lower effort. Finally, all participants significantly reduced effort post-exposure. This work demonstrates the feasibility of using exoskeletons as biofeedback systems to improve posture, paving the path for applications in musculoskeletal disorders prevention.

In the past decades, active exoskeletons have been used to reduce human effort, whether it be to allow neurorehabilitation[1] or to assist workers while performing demanding tasks[2]. A potential limit of this approach is that it does not aim at guiding the user to learn new, more efficient and ergonomic postures. In fact, simply reducing the amount of human effort required to perform a task could even lead to the adoption of detrimental postures, which may shift the risk of developing musculoskeletal disorders (MSD) from a joint to another[3,4]. The ability to simultaneously assist movements and teach efficient and protective postures to the exoskeleton's user would be critical to reduce the long-term risk of MSD. In the present study, we investigate a method using active exoskeletons to implicitly guide users toward ergonomic postures in tasks where multiple postures may be used, as it is usually the case in daily-life.

Traditional biofeedback approaches to inform humans about the risks associated with their posture date back to the 1960s. They have mainly employed additional sensory feedback streams conveying information related to the behavior or real-time efforts supported by the user, both for MSD prevention and neurorehabilitation[5–7]. Such biofeedback relies on the monitoring of physiological variables through sensors, the most common of which are electroencephalographs (EEG), for clinical applications[5,8], electromyographs (EMG)[9], motion capture systems[7,10–13], force plates[14], and heart rate sensors[15,16]. Using these signals, one can estimate in real-time variables, such as patterns of neural activity[5], joint torques[17–19], and posture[12]. Then, biofeedback consists of generating a sensory stimulation to inform the user with regard to their own behavior, and to trigger warnings in case they exceed a risk or effort threshold. The most common sensory stimulation uses visual[20], auditive[20,21], and somatosensory (usually through vibrators) cues[11,22–25], or a combination thereof[10,26]. These traditional approaches have been proven to be efficient in modifying human behavior and promoting the adoption of better postures. However, they present two major limitations: (i) they do not allow for guiding users toward adopting a precise posture, as this would require a large number of stimulating devices (typically several per joint), and (ii) the need for explicitly interpreting the additional sensory signals of multiple stimulating devices could result in a neural resource allocation problem[27]. This concept was recently introduced to discuss the limited resources available to the brain to simultaneously control the movements of natural limbs and supernumerary limbs (SLs), which provide additional degrees of freedom (DoFs) to the human body. This control requires both to process additional sensory signals associated with SLs and generate relevant motor commands for SLs. In this context, the finiteness of human neural resources may lead to decreased natural limb control performance when interacting with SLs. This concept also applies to

[1]Université Paris-Saclay, Inria, CIAMS, Gif-sur-Yvette, France. [2]Enchanted Tools, Paris, France. [3]LURPA, ENS Paris-Saclay, Université Paris-Saclay, Gif-sur-Yvette, France. [4]Université Rennes 2, M2S, Inria, Rennes, France. [5]Bioengineering department, Imperial College of Science, Technology and Medicine, London, UK. ✉e-mail: d.verdel@imperial.ac.uk

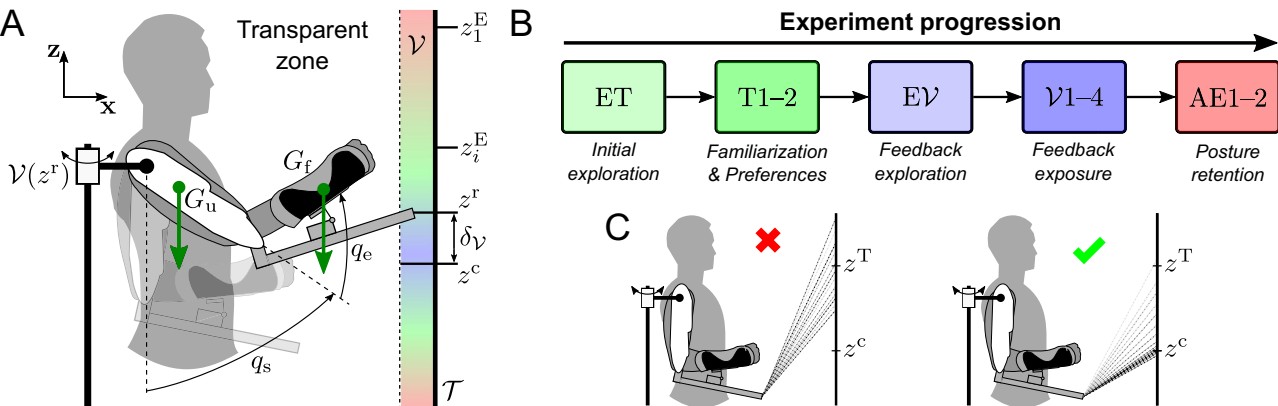

**Fig. 1 | Human-exoskeleton manifold reaching experiment with *gravito-vibratory landscape*. A** Participants performed reach-to-hold movements with shoulder and elbow flexion/extension movements (with angles $q_s$ and $q_e$ respectively) towards a vertical bar $\mathcal{T}$, implying a target redundancy[39]. While maintaining their final posture, participants were subjected to (i) gravity (green arrows), due to the weight of their upper arm at $G_u$ and forearm at $G_f$, and (ii) a vibration $\mathcal{V}$, applied by the exoskeleton along **y**. The vibration evolved as a function of the absolute distance between the final height $z^t$ and the center (i.e., cancellation point) of the vibratory landscape $z^c$, noted $\delta_\mathcal{V}$. The center $z^c$ was set 20 cm above or below the preferred endpoint height of each participant, resulting in two groups. The preferred endpoint height was the median height reached across 40 trials, performed during the transparent block (T). **B** Structure of the experiment. The protocol started with

an exploration block in transparent mode (ET), where participants reached towards ten predefined heights equally spread along the vertical bar $z_i^E$, with $i \in [[1, 10]]$, and held their posture for 3 s. Then, they performed two blocks of reach-to-hold movements in transparent mode (T). These were followed by an exploration block with the vibration (E$\mathcal{V}$) and four blocks with the vibration ($\mathcal{V}$). Finally, two blocks were performed in transparent mode to analyze possible after-effects (AE).
**C** Possible outcomes of the interaction with the landscape, illustrated with successive trajectories of the exoskeleton's end-effector— represented as straight lines for the sake of simplicity—from the firsts (light gray) to the lasts (black) trials. Left: the participant fails to converge to $z^c$ and remains in the vicinity of $z^T$. Right: the participant gradually converges toward $z^c$.

biofeedback methods, as they generate additional sensory stimulation to those normally processed by the CNS for a given task. Such stimulation creates a risk of diverting neural resources from their usual role, inducing an increased cognitive cost to maintain performance.

We propose to address these limitations by developing an alternative to traditional biofeedback approaches. Specifically, we propose to use an active exoskeleton as an efficient way to (i) measure posture—which requires to identify joints misalignment to be accurate[28,29]—and (ii) generate sensorimotor stimulation modifying the posture-effort relationship to trigger implicit motor adaptations. This way, the exoskeleton performs both the monitoring and stimulation components of biofeedback, removing the need for additional devices. We argue that the sensorimotor stimulation provided by the exoskeleton should not be limited to simple guidance methods, for instance, previously used to teach movements[30]. Instead, we utilize the exoskeleton to introduce task-relevant variability through a haptic signal, aiming to promote central nervous system (CNS) adaptation by generating motor errors. In other words, if the human does not adapt to the feedback provided by the exoskeleton, they will not be able to do the task. This task-relevant characteristic is critical as it is also known that task-irrelevant variability, i.e., not hampering successful task completion, may be ignored by the human sensorimotor system[31,32], which could prevent any meaningful adaptation. This is likely due to the learning of an internal model[33] of new motor behaviors through the memorization[34] and prediction[35] of movement errors, which needs rich and varied input data to robustly control movements[36]. In addition to removing the need for separate and numerous sensory and stimulation devices, such an approach presents the advantage of being highly flexible. Theoretically, any posture-stimulation mapping can be designed, meaning users could be guided toward any reachable posture, through a simple combination of usual visual information and a stimulation of the somatosensory system with forces directly relevant to the task. Such forces are known to be optimizable by the CNS even when unpredictable[37,38], thereby addressing the risk of a neural resource allocation problem.

This approach needs to be experimentally tested so as to conclude on its feasibility and usability. Here, we focus on the general feasibility of the method, without selecting targeted postures based on a specific ergonomic score. To remain as general as possible, we investigate two cases: one where the targeted posture is better and one where it is worse than the participant's

preferred one. These investigations are conducted using a redundant reach-to-hold experiment with the following main characteristics (see Figs. 1, 2): (i) the target of the reaching movement was a long vertical rectangle, allowing exploration (ET) of various final postures through target redundancy[39], (ii) the movements were in a parasagittal plane, where gravity-related efforts vary significantly, which allowed to differentiate between ergonomic (with lower effort) and non-ergonomic (with higher effort) postures, (iii) participants were exposed to a haptic stimulation shaped as lateral vibrations applied by the robot at the end of the reaching phase to push them outside of the target, thereby ensuring its task-relevance (vertical vibrations would not prevent the successful completion of the task), and (iv) the vibrations' amplitude exhibited a minimum (null vibration) that was chosen either above or under the posture each participant nominally preferred. When the minimum of the vibrations was above the preferred height, three ergonomic features were impacted: (i) gravity-related efforts were increased (see Fig. 2A); (ii) shoulder and elbow angles were closer to their upper-limits[40,41]; and (iii) blood-flow was reduced due to gravity and compression of the tendons in the shoulder, especially when $z^c$ was above the shoulder[42,43]. Conversely, when the minimum of the vibrations was below the preferred height, it mechanically led to an improvement of these three ergonomic criteria. In the rest of the present paper, the combined effects of gravity-related efforts and the height-dependent vibrations applied by the exoskeleton are referred to as the *gravito-vibratory landscape*, the first being ubiquitous and natural, while the other is artificially introduced for sensorimotor guidance. The investigations focus on verifying two hypotheses: (i) whether participants succeed or fail to adapt by adopting a posture minimizing the intensity of vibrations and/or of gravity-related efforts, (ii) whether they can learn and retain adapted postures different from their nominally preferred one, which would be critical for ergonomics improvement (see Fig. 1C).

## Results

The experiment included three main conditions. First, participants familiarized themselves with the task by performing an ET block, and two blocks of movements with the exoskeleton in transparent mode[44]. During the ET block, participants reached and held ten predefined postures equally spread across the workspace, which (i) allowed them to build a prior of the efforts

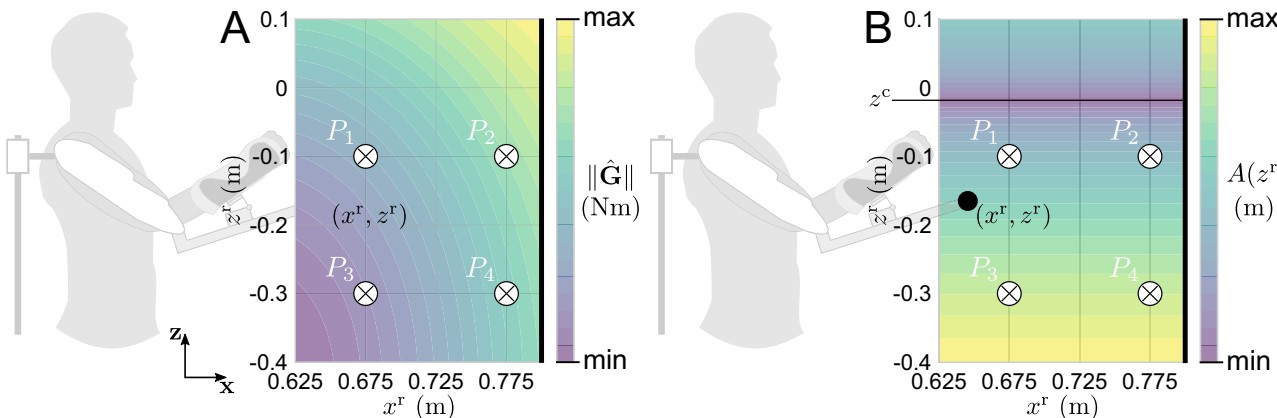

**Fig. 2 | Illustration of the evolution of the two components of the *gravito-vibratory landscape*.** Landscapes are here illustrated for the zone where most movements ended, which is from 0.4 m below the shoulder to 0.1 m above the shoulder (**z** axis). On the **x** axis, it was computed from 0.625 to 0.8 m. This interval on the **x** axis represents the range of valid reaching depths, in which (i) the height-dependent vibration was activated in the EV and V blocks, and (ii) the task could be validated. $P_1$–$P_4$ represent how different end-effector positions can largely affect the

effort requested to hold the posture. The black vertical bar on the right represents the target displayed to the participants. **A** Gravity landscape, with $\|\hat{\mathbf{G}}\|$ the norm of the gravity torques, including the elbow and shoulder. **B**. Vibratory landscape, with $A(z^r)$ the height-dependent amplitude of the vibration. In this situation, the gradients of both components are of opposite signs, implying that holding a higher posture increases gravity-related efforts but decreases the vibrations magnitude.

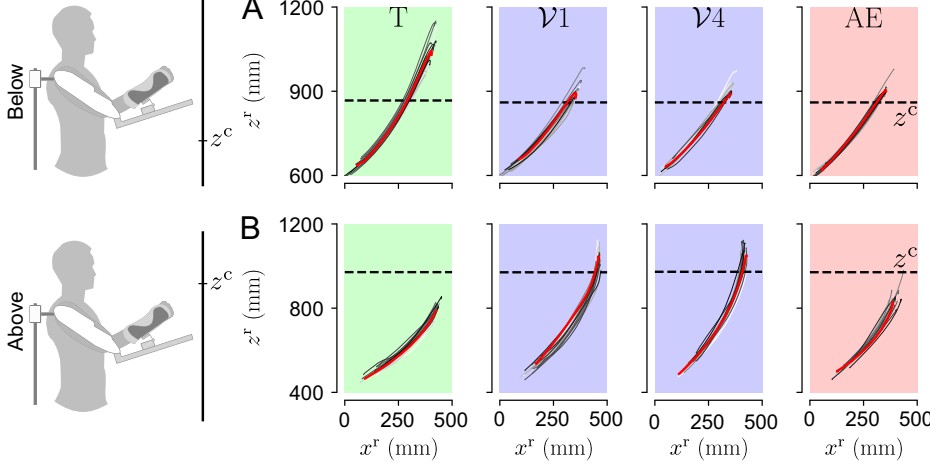

**Fig. 3 | Reaching trajectories of representative participants for each group through the experiment.** The red curves represent the trajectories with the median final height ($z^r$) during the transparent (T), first and last vibration ($V_1$ and $V_4$) and after-effects (AE) blocks. Horizontal dashed line represents the height at which the amplitude of the vibrations was null. **A** *Below* group. **B** *Above* group.

and comfort associated with postures across the target space, and (ii) allowed consistency with the $V$ conditions that are preceded by the same ET to build a prior regarding the task with added task-relevant variability. The two transparent blocks allowed us to extract the preferred final posture of each participant by extracting the median reached height of the 40 trials performed in this mode. Then, they performed four blocks of reach-to-hold trials with a lateral vibration applied by the exoskeleton ($V_1$–$V_4$) at the final posture, resulting in a *gravito-vibratory landscape* (see Fig. 1B). The amplitude of the vibration was dependent on the endpoint's height of each participant (see Eq. (1) in Methods), with a null amplitude at the center of the landscape $z^c$ placed at ± 20 cm of the median preferred height of each participant. The assumption behind our method being that participants can be implicitly guided toward a specific posture with task-relevant haptic perturbations, no visual feedback is provided regarding the position of $z^c$ in the target space. For half of the participants, who naturally selected postures low in the target space, $z^c$ was chosen *above* their preferred endpoint height, and it was chosen *below* for the other half of the participants, who naturally selected postures close to the height of their shoulder. Finally, participants performed two blocks with the transparent exoskeleton, allowing us to analyze the after-effects (AE) of exposure to the gravito-vibratory landscape. Importantly, both these components (gravity-related efforts and vibrations)

evolved as a function of the final posture to hold ($x^r$, $z^r$) and exhibited a minimum in the reachable space. Their gradients were of opposite sign for the *above* group, i.e., reducing the vibrations increased gravity-related efforts, whereas they had the same sign for the *below* group, i.e., reducing vibrations decreased gravity-related efforts. This separation into two groups allows us to test our method's ability to implicitly guide users toward a specific posture in two contrasted contexts. First, in the case of the *below* group, the vibration cancels at a more ergonomic final posture than the participant's preferred one. Conversely, for the *above* group, the vibration cancels at an ergonomically worse posture.

We first qualitatively analyzed the ET and optimization of the final posture using representative trajectories throughout the experiment for each group, as summarized in Fig. 3.

Interestingly, Fig. 3 suggests that both groups adapted their reach-to-hold strategy to minimize the vibrations applied by the exoskeleton at their final posture. During the AE blocks, the participant from the *above* group seems to have moved back to their initial strategy, while the participant from the *below* group seems to have changed their strategy, retaining their final posture from the blocks with the gravito-vibratory landscape. To better understand how participants explore the landscape, we then extracted "navigation maps" representing the influence of the selected postures on the

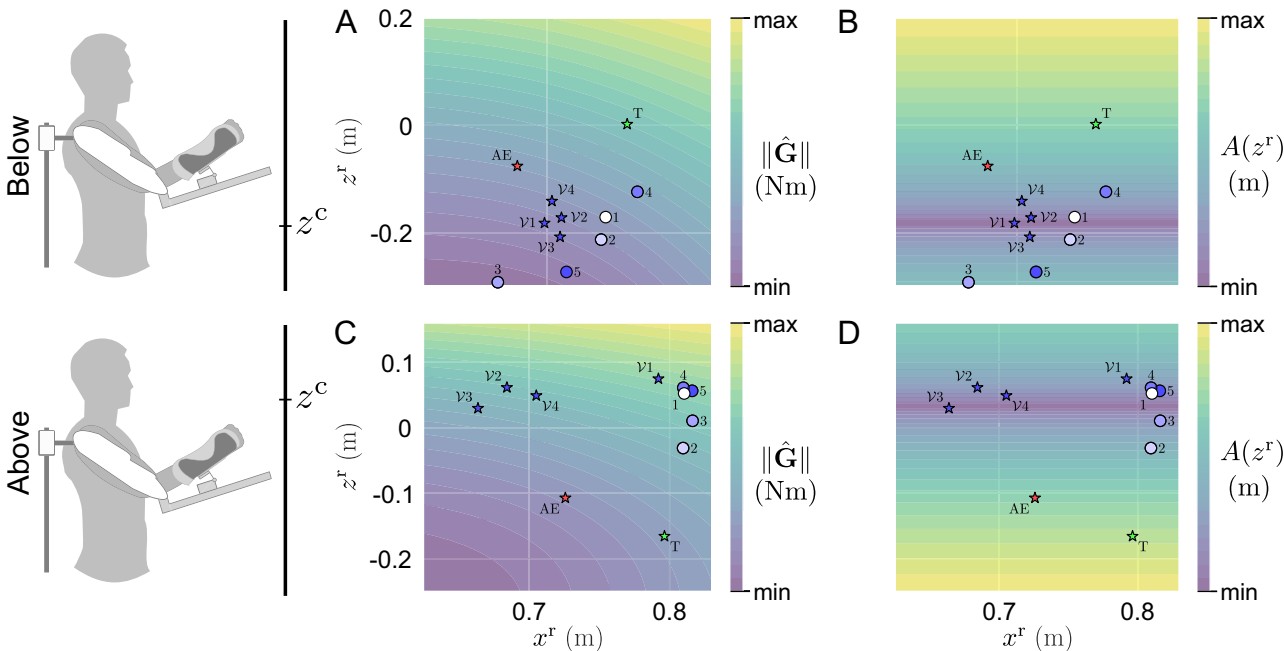

**Fig. 4 | Navigation maps of two representative participants of the *above* and *below* groups.** The stars represent the median final position in each block. The disks represent the first five trials in the first vibratory $\mathcal{V}_1$ blocks, showing the initial exploration of the gravito-vibratory landscape. **A, C** Representative navigation of the gravity component $\|\hat{\mathbf{G}}\|$ for the *below* and *above* group, respectively. **B, D** Representative navigation of the vibratory component $A(z^r)$ for the *below* and *above* group, respectively.

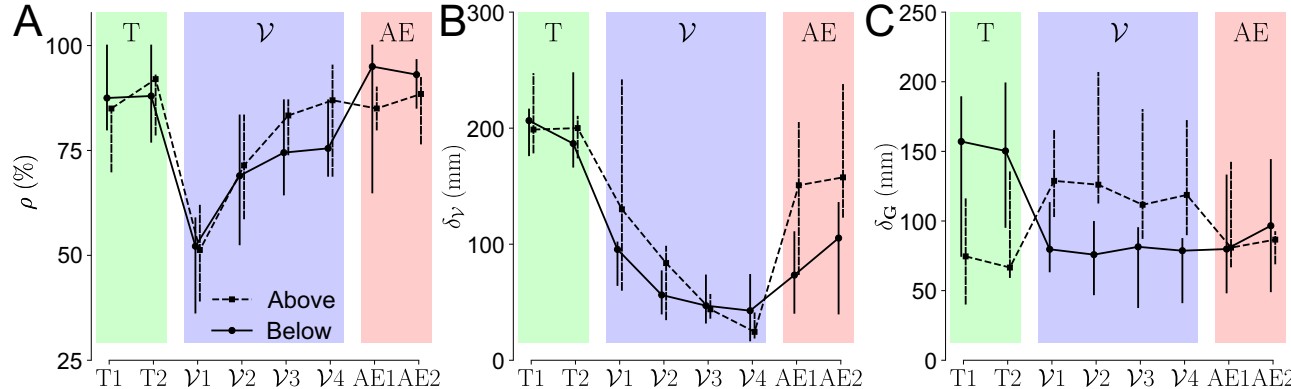

**Fig. 5 | Adaptation of the main interest parameters for each group throughout the experiment.** Transparent (T1–2), vibration ($\mathcal{V}$1–4) and after-effects (AE1–2) trials are in green, blue and red shaded areas, respectively. The *above* group is represented with dashed lines and the *below* group with solid lines. Markers represent the median across participants and vertical bars represent the confidence intervals. **A** Success rate $\rho$. **B** Distance to the minimum of the vibratory landscape $\delta_\mathcal{V}$. **C** Distance to the minimum of the vibratory landscape $\delta_\mathbf{G}$.

intensity of each component through the experiment. These analyses are summarized in Fig. 4.

The navigation maps for the gravity component of the landscape suggest that both groups tended to minimize effort by reducing their arm extension when the landscape was activated. For both the *above* and *below* groups, the median final posture in AE seems to retain this reduced arm extension. The navigation of the vibratory component by both representative participants suggests that they tried to minimize the intensity of the vibration. Interestingly, they seem to both succeed in doing this through ET.

Following these qualitative analyses, we quantified the adaptation of the final posture with respect to three main variables: the success rate in each block $\rho$, and the evolutions of $\delta_\mathcal{V}$ (see Eq. (2)) and $\delta_\mathbf{G}$ (see Eq. (4)) throughout the experiment. These two distances, $\delta_\mathcal{V}$ and $\delta_\mathbf{G}$, represent how close to the optimum the final posture is with respect to each landscape component. Block-wise analyses of these parameters are summarized in Fig. 5.

The evolution of the success rate through the blocks $\mathcal{V}_1$–$\mathcal{V}_4$ suggests that participants gradually learned to perform the task, as illustrated by a significant main effect of the block for both the *above* and the *below* group ($\chi^2 = 3498$, $p < 0.001$). Specifically, $\mathcal{V}_1$ exhibits a drop in the success rate $\rho$ from above 85% to around 50% when compared to the transparent blocks, which was confirmed to be significant for both groups (in both cases: $p < 0.001$, Cohen's $D > 3.27$). Then, $\rho$ gradually increases with learning, as illustrated by the significant differences between $\mathcal{V}_1$ and $\mathcal{V}_4$ for both groups (in both cases: $p < 0.001$, Cohen's $D > 2.48$). Interestingly, the performance of the *above* group in $\mathcal{V}_4$ was comparable to their performance in the transparent blocks, while the performance of the *below* group was still significantly lower ($p < 0.001$, Cohen's $D = 0.984$). Finally, the success rate in the AE blocks was similar to the transparent blocks for both groups, which was significantly better than in $\mathcal{V}_4$ (in both cases: $p < 0.001$, Cohen's $D > 0.47$), with a larger and clearer improvement for the *below* group. In

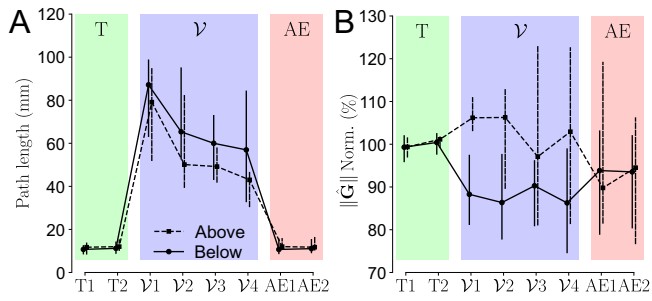

**Fig. 6 | Effects of the gravito-vibratory landscape on the human effort throughout the experiment.** Transparent (T1–2), vibration ($\mathcal{V}$1–4) and after-effects (AE1–2) trials are in green, blue and red shaded areas respectively. The *above* group is represented with dashed lines and the *below* group with solid lines. Markers represent the median across participants and vertical bars represent the confidence intervals. **A** Total lateral distance traveled by the hand while maintaining the final posture in the landscape. **B** Normalized gravity-related efforts averaged while maintaining it.

summary, the evolution of the success rate throughout the blocks clearly indicates that participants learned how to successfully perform the task. However, this does not allow us to conclude regarding their strategy to succeed. In particular, they could either resist the vibratory landscape through a modulation of their impedance[45], or learn to minimize its intensity through exploring the target space.

To investigate these two possibilities, we analyzed the evolution of $\delta_{\mathcal{V}}$, i.e. the distance to the null vibration height, through the blocks $\mathcal{V}_1$–$\mathcal{V}_4$. For both groups, there was a clear and significant decreasing trend in this distance with practice ($\chi^2 = 463$, $p < 0.001$), which seemed steeper for the *below* group. Specifically, $\delta_{\mathcal{V}}$ was significantly smaller in $\mathcal{V}_1$ than in the transparent blocks (in both cases: $p < 0.001$, Cohen's $D > 1.57$). Furthermore, $\delta_{\mathcal{V}}$ tended to decrease between $\mathcal{V}_1$ and $\mathcal{V}_4$, which was significant for the *above* group ($p < 0.001$, Cohen's $D = 0.603$). During the AE blocks, participants tended to retain some of the adaptations induced by the vibratory landscape, as illustrated by the significant decrease in $\delta_{\mathcal{V}}$ when compared to the transparent blocks (in both cases: $p < 0.001$, Cohen's $D > 0.45$). However, this retention was not complete as $\delta_{\mathcal{V}}$ significantly increased during AE compared to $\mathcal{V}_4$ (in both cases: $p < 0.001$, Cohen's $D > 0.72$). Remarkably, the *below* group showed a higher retention than the *above* group during AE, as illustrated by significantly lower $\delta_{\mathcal{V}}$ ($p < 0.001$, Cohen's $D = 0.914$). In sum, participants gradually learned to minimize the effects of the vibratory landscape through the four performed blocks. The *above* group did not retain much of the learned optimized posture, while the *below* group tended to remain closer to their behavior in $\mathcal{V}_4$. Furthermore, the *below* group adapted their behavior to the vibratory landscape faster than the *above* group, although both groups converged to comparable levels of minimization by the end of the four blocks. These important differences between groups suggest that the gravity component of the landscape, in particular, whether the gradients of both landscape components had the same sign, could play a critical role in the adaptability and retention of postures by participants.

Thus, we analyzed the evolution of $\delta_{\mathbf{G}}$, i.e., the distance to the minimum of $\| \hat{\mathbf{G}} \|$ (defined as the norm of the gravity-related efforts, see Eq. (3)) within the target space, through the blocks $\mathcal{V}_1$–$\mathcal{V}_4$. These analyses exhibited a significant main effect of the block on the participants' final posture ($\chi^2 = 731$, $p < 0.001$). As expected due to how they were formed, there was a preexisting difference in $\delta_{\mathbf{G}}$ between the *above* group, with lower $\delta_{\mathbf{G}}$, and the *below* group, with higher $\delta_{\mathbf{G}}$ ($p < 0.001$, Cohen's $D = 0.859$). Then, as soon as in $\mathcal{V}_1$, this difference in distance from the minimum gravity effort was reversed, and the *below* group exhibited significantly lower $\delta_{\mathbf{G}}$ than the *above* group ($p < 0.001$, Cohen's $D = 1.194$). Incidentally, the *above* group moved significantly further away from the minimum of efforts, while the *below* group moved closer to it when compared to transparent blocks (in both cases: $p < 0.001$, Cohen's $D > 0.95$). This effect remained present until the end of the four blocks within the gravito-vibratory landscape, as

illustrated by significantly higher $\delta_{\mathbf{G}}$ for the *above* than for the *below* group in $\mathcal{V}_4$ ($p < 0.001$, Cohen's $D = 1.277$). Through the AE blocks, the *above* group moved back to their $\delta_{\mathbf{G}}$ of the transparent blocks (no significant difference), which was reminiscent of observations for $\delta_{\mathcal{V}}$. The *below* group partially retained the novel final posture they learned through the interaction with the gravito-vibratory landscape, as illustrated by the small difference between $\mathcal{V}_4$ and AE ($p = 0.029$, Cohen's $D = 0.31$, small effect) and by a large and significant difference between the transparent blocks and AE ($p < 0.001$, Cohen's $D = 0.978$). In sum, the similar signs of gradients of the landscape's components led the *below* group to retain the posture to which they were guided, whereas, as soon as the vibration was removed, the *above* group moved to a less effortful posture.

It should be noted that the gravito-vibratory landscape evolves non-linearly in the task space (see Fig. 2). Therefore, although we showed how the posture was retained or not depending on the distance to the minimum effort and minimum vibration, this is not sufficient to conclude regarding the final posture retained by the participants. To quantify the effects of the gravito-vibratory landscape on humans more precisely, we computed the total lateral distance traveled by the hand after reaching the posture and the average normalized gravity effort while maintaining it. The normalization of gravity-related efforts was done with respect to the median $\| \hat{\mathbf{G}} \|$ in the transparent blocks, using Eq. (3). The evolution of these two metrics through the experiment is summarized in Fig. 6.

The analysis of the total lateral displacement of the hand allows to verify how much of the vibration the participants were able to compensate using muscle effort. There was a significant main effect of the block on the participants' hand lateral movements ($\chi^2 = 2039$, $p < 0.001$). Specifically, there was only residual lateral movement in the transparent block, as permitted by passive joints in the human-exoskeleton connection (see Methods). In $\mathcal{V}_1$, the lateral distance traveled by the hand dramatically increased for both groups (in both cases: $p < 0.001$, Cohen's $D > 3.36$), consistently with observations of $\delta_{\mathcal{V}}$ suggesting that they did not find the minimum of the vibration landscape yet. The lateral movements then decreased with practice to reach a minimum in $\mathcal{V}_4$, which was significantly lower than in $\mathcal{V}_1$ (in both cases: $p < 0.001$, Cohen's $D > 1.29$). Interestingly, the *above* group's hands moved significantly less than the *below* group's in $\mathcal{V}_4$ ($p < 0.001$, Cohen's $D = 0.553$), suggesting that their posture allowed them to better compensate for the vibrations as no difference was observed in $\delta_{\mathcal{V}}$ in this block. For both groups, this minimum was still significantly higher than the residual movements observed in both the transparent and AE blocks (in all cases: $p < 0.001$, Cohen's $D > 1.75$). In sum, participants never completely compensated for the vibrations, which suggests that they only compensated for as much vibration as necessary to remain in the target, consistently with a task-dependent minimum intervention hypothesis[46]. This optimal use of the width of the target allows us to finalize the analysis of the participant's behavior with respect to the vibratory component of the landscape.

However, it does not allow to provide a definitive answer regarding the optimization of the gravity component of the landscape. Therefore, we analyzed the evolution of the normalized gravity-related efforts to maintain the final posture throughout the experiment, revealing a significant main effect of the block on the effort provided by the participants ($\chi^2 = 164$, $p < 0.001$). In $\mathcal{V}_1$, the effort slightly increased for the *above* group ($p = 0.015$, Cohen's $D = 0.353$, small effect) and decreased for the *below* group ($p < 0.001$, Cohen's $D = 0.494$), resulting in divergent effects of the landscape on the effort expended by both groups ($p < 0.001$, Cohen's $D = 0.897$). There was no clear adaptation of the gravity-related efforts throughout blocks $\mathcal{V}_1$ to $\mathcal{V}_4$, consistently with results on $\delta_{\mathbf{G}}$. Interestingly, the *above* group moved to a less effortful posture in AE when compared to $\mathcal{V}_4$ ($p < 0.001$, Cohen's $D = 0.476$), while no significant difference was observed for the *below* group. Finally, in AE, both groups adopted postures that reduced gravity-related efforts when compared to transparent blocks (in both cases: $p < 0.022$, Cohen's $D > 0.26$, small effects). In sum, both groups adapted to the gravito-vibratory landscape so as to make task-relevant vibrations smaller than the width of the target, which was observed whether it was beneficial or detrimental to the effort they had to provide. After exposure, despite variability in

the results, we could observe that both groups improved their posture in the sense that they reduced the gravity-related efforts.

## Discussion

We investigated whether task-disturbing somatosensory biofeedback applied by a robotic exoskeleton is a feasible method to guide users toward more ergonomic postures, thereby preventing the development of MSD. When performing tasks with multiple solutions, humans can sometimes adopt non-ergonomic postures, needlessly putting themselves at risk of MSD. While this can be due to a number of psychosocial factors, including stress and time constraints[47] or the design of work stations[48,49], it can also sometimes be the consequence of bad habits and a lack of knowledge about ergonomics[50]. It can also be related to the incapacity of our CNS to consider the long-term effects of non-ergonomic postures. In this context, our idea was to design a method that leverages the spontaneous minimization of task-relevant variability in human motor control to teach novel postures to exoskeleton users through posture-dependent mechanical stimulation. Specifically, we designed a *gravito-vibratory landscape* that varied with the user's posture and exhibited a minimum for each component within the task target space. The main hypothesis validated by the experiment was twofold. First, minimizing the landscape intensity through trial-by-trial gradient descent was possible, allowing participants to find an ergonomic posture that traded off the costs of vibrations and gravity to perform the task. Second, participants could retain the novel posture discovered through ET if it reduced the effort compared to their preferred posture before exposure, illustrating the benefits of using an exoskeleton to guide users toward more ergonomic postures.

At the population level, our results clearly show that participants gradually learned to minimize the effects of the vibratory component of the landscape, until the remaining vibrations were sufficiently small to comply with the task constraints. Our vibratory landscape, generating large task-relevant variability, could incentivize the CNS to perform a "gradient descent" through trials, thereby minimizing the mechanical effects of the disturbance. The residual vibrations that were not compensated by participants after four blocks of exposure further suggest that the task-relevance of the perturbation is critical for them to adapt, as predicted by the minimum intervention principle[31,32,46]. Our results also showed that participants largely adapted to the gravity component of the landscape within the first block of exposure to vibrations, contrasting with the longer adaptation to the vibratory component. This difference in the time scale of the adaptation is likely due to the omnipresence of gravity in our environment. Specifically, the CNS is thought to plan movements leveraging gravity-related efforts[51], as also demonstrated for primates[52] and when interacting with unusual gravity-like torques applied by an exoskeleton[53]. This rapid change in the gravity component of the landscape is particularly important for preventing MSD, as gravity-related efforts are known to play a significant role in their development. Specifically, ergonomic postures are those that minimize fatigue, which often corresponds to minimizing gravity in work conditions. This applies when carrying loads[54,55], using screens[56,57], or when performing highly constrained tasks, such as laparoscopic surgery[58]. Yet, at the group level, roughly half of our participants naturally selected postures close to or above the shoulder height (*below* group), which implies relatively high, unnecessary, gravity-related effort to be maintained. Through the exposure to the gravito-vibratory landscape, we showed that this *below* group rapidly decreased gravity-related efforts in reaction to the task-relevant vibrations, thereby improving their posture. More surprisingly, the *above* group naturally and rapidly increased gravity-related efforts in an attempt to reduce vibrations, showing the higher priority assigned to the successful task completion. These results demonstrate that applying error-generating sensorimotor stimulation to implicitly guide a worker wearing an exoskeleton toward novel postures is feasible, and that adaptations happen within a relatively small number of trials.

We show contrasted results in the retention of the posture suggested through the gravito-vibratory landscape. When the novel final posture obtained through ET induced higher gravity-related efforts (*above* group), it was clearly not retained by the participants. Conversely, when the minimum of vibrations induced a reduction in these efforts (*below* the group average), the posture tended to be retained, thereby canceling preexisting differences between groups in the distance to the posture that minimized gravity-related efforts. In both cases, participants showed an improvement of the ergonomics of their preferred posture, as illustrated by the lower norm of gravity-related efforts. This suggests that forcing users to explore a variety of postures allows them to better minimize the efforts associated with a task. In addition to minimizing task errors, this finding may be explained in part by the natural movement variability that allows users to iteratively explore different final postures. There is compelling evidence in movement neuroscience that human motor learning is more efficient when they can explore the task space[59,60], allowing them to optimize motor behaviors. In fact, when asked to perform a novel task, individuals with high levels of task-relevant variability tend to learn faster than those exhibiting lower levels of variability[61]. Therefore, the freedom given to participant in selecting their final posture may contribute to their retention and improvement of final posture. This illustrates the potential benefits of leveraging this fundamental mechanism of human movement control for ergonomics. However promising, these results would need to be consolidated through longitudinal studies, allowing a longer and repeated exposure as well as long-term retention analyses, which were beyond the scope of the present work. Those studies, potentially involving repeated exposure to the gravito-vibratory landscape over several sessions, would allow us to investigate whether the posture adaptations that we report can be sustained over long periods. Furthermore, extending those ideas to whole-body exoskeletons in load-carrying tasks could also be relevant, possibly with more accurate techniques than anthropometric tables, for example by identifying the human limbs dynamics from interaction efforts[28,62].

Although systematic comparisons between our method and traditional biofeedback techniques will be needed to assess differences in performance and resulting user benefits, several functional findings can already be made. When compared to traditional biofeedback methods, our approach solves several important issues. On the sensing side, as it relies on an exoskeleton to measure posture, it does not require any other sensors than those embedded in the robot, which drastically limits the installation and calibration complexity and the required signal processing. This is particularly practical when compared to approaches based on electromyographic measures of muscle activity[9,63], or motion capture[7,10–13]. However, this means that our method requires knowledge of task-related ergonomic postures, which may not be necessary with other approaches that record the autonomous nervous system to infer effort[15,16]. This problem being common to a variety of ergonomic assessments and biofeedback approaches, a large number of tools to address it are available[64], including posture-based scales[65,66] and more advanced assessments explicitly accounting for effort[41]. On the actuation side, our method only stimulates the sensory streams already involved in performing the task, in particular the somatosensory system at the level of the human-exoskeleton interfaces, which prevents the risk of overloading the brain with sensory information[27].

This stimulation can be easily applied to any task or posture, without increasing the complexity of the setup, unlike other biofeedback systems. In the present paper, we have only investigated the case of a unimanual task, but we argue that bimanual tasks could also be treated using our method whenever the user can be equipped with a bilateral active exoskeleton. In practice, the implementation would depend on whether the two hands are mechanically coupled, e.g., when carrying a large box, or not, e.g., when holding one object in each hand. In the first case, the haptic perturbation could be common, depending on the pose of the manipulated object. In the other case, signals provided on each side can be independent. For comparison, when using vibrators as is commonly done, at least one actuator would be required for each monitored joint to induce a specific posture, entailing a system complexity dependent on the number of joints. The need for an actuated robotic device may be seen as a downside due to their cost and the difficulty of their use compared to vibrators, speakers or visual feedback devices (e.g., mixed reality glasses). However, given that

exoskeletons are already envisioned as assistive devices for the prevention of MSD[67,68], the proposed approach would therefore require minimal investment when added on top of existing in-field setups.

In case further long-term studies are conclusive, our method could be applied to an industrial scenario, such as an automotive assembly task with multiple possible postures, using a simple procedure. The first step would be to assess what are the best postures that can be used to perform the task, for instance using existing ergonomic tools[64]. A vibratory landscape with a null intensity near these ideal postures can be implemented, resulting in bad postures being highly penalized. Then, both novice and expert workers can be exposed to this landscape during dummy training sessions, outside of production lines as performance is directly impacted during learning. The exposure sessions can be repeated on a regular basis so as to remind the CNS of the best postures to adopt, thereby preventing the appearance of MSD. Finally, the haptic stimulation can be coupled with an assistance helping to perform the task, for instance, by compensating gravity[28,69] or with adaptive behaviors[70–72], allowing to further reduce the cost of ergonomic postures and paving the path for ergonomic-aware assistance.

## Methods
### Participants, materials, and data processing
**Participants**. The experimental protocol was approved by the Université Paris-Saclay's ethics committee for research (CER-Paris-Saclay-2021-048). A total of $N = 15$ (8 in the *below* and 7 in the *above* group) healthy, right-handed, and naive participants (11 males, and four females) participated in the experiment. Their anthropometric characteristics were: age $23.7 \pm 3.3$ years old, height $1.79 \pm 0.09$ m, and weight $68.80 \pm 7.12$ kg. Each participant was informed about the experiment and signed an informed consent form before participating in it.

**Upper-limb kinematics**. We recorded the participant's upper-limb movements at 100 Hz using an optoelectronic motion capture system (10 Oqus 500+ infrared cameras, Qualisys, Gothenburg, Sweden). Each participant was equipped with 10 mm reflective markers allowing to measure the positions of their upper arm and forearm, as well as the position of the exoskeleton in real-time (see Fig. 1A for representative positions of the markers). Note that the markers placed on the participants were mainly used to identify the human-exoskeleton joints misalignment in a simple way. Markers were also used to estimate the position of the exoskeleton's end-effector in the task space, thereby removing pose estimation errors due to the robot's flexibility. In practice, both this identification and estimation can be performed using other information readily available in the exoskeleton, such as interaction forces[28].

**ABLE exoskeleton**. We used a highly backdriveable robotic upper-limb exoskeleton called *ABLE*[73,74] to physically interact with the participants (see Fig. 1A). This exoskeleton includes four active DoFs, three of which are at the shoulder level (flexion/extension, abduction/adduction, internal/external rotation), and one at the elbow (flexion/extension). This version of ABLE includes ergonomic human-robot physical interfaces to connect the arm and forearm of the participant to the robot[75,76], which have been shown to reduce unwanted interaction efforts and increase comfort. Furthermore, the exoskeleton is equipped with two force-torque (FT) sensors (1010 digital FT, ATI, sample rate 1 kHz) placed at each human-exoskeleton interface point, which allowed to measure interaction forces to control the robot and improve transparency[44].

The exoskeleton was mostly designed to be as transparent as possible, so as to follow human movements with a minimal impact on it. The transparent controller relied on the identification and compensation of the exoskeleton's weight and friction and on the minimization of the interaction efforts using a proportional-integral correction, that was shown to be efficient in previous works[44,77]. In a certain series of trials, when the end-effector of the exoskeleton entered the $\mathcal{V}$ zone (see Fig. 1B), the motor providing the internal/external rotation of the exoskeleton's shoulder applied motor

perturbations depending on the human posture to generate a vibratory landscape as described below.

**Gravito-vibratory landscape**. The effort landscape, at the end of reach-to-hold movements, generated by the exoskeleton and the dynamics of the human arm was composed of two main components: (i) the vibratory landscape imposed by the exoskeleton, and (ii) the gravity-related efforts to hold the posture.

First, we designed a lateral vibration with a varying amplitude, aimed at perturbing participants while they held their final posture. This task-relevant vibration was defined as follows,

$$\mathcal{V}(z^{\mathrm{r}}) = A(z^{\mathrm{r}}) \sin(2\pi f t), \text{ where } A(z^{\mathrm{r}}) = A_0 \sqrt{\frac{|z^{\mathrm{c}} - z^{\mathrm{r}}|}{\beta + |z^{\mathrm{c}}|}} \quad (1)$$

where $A(z^{\mathrm{r}})$ is the varying amplitude, $A_0 = 9°$ and $f = 8$ Hz parametrize the amplitude and frequency of the lateral vibration imposed by the exoskeleton, $z^{\mathrm{r}}$ is the vertical coordinate of the exoskeleton's end-effector in the task space, $z^{\mathrm{c}}$ is the "center" of the landscape, i.e., where the amplitude of the vibration is null, and $\beta = 50$ was chosen to provide a smooth but sufficiently steep increase of the force field as a function of the distance between $z^{\mathrm{r}}$ and $z^{\mathrm{c}}$. From Eq. (1), one can define the distance to the center of the landscape, which minimizes the effort to provide to laterally stabilize the robot, as follows,

$$\delta_{\mathcal{V}} = |z^{\mathrm{r}} - z^{\mathrm{c}}| \quad (2)$$

The computation of this distance at the end of each reach-to-hold movement performed by the participant served as one of the main metrics to quantify their adaptation to the vibratory landscape and analyze whether they adopted more ergonomic postures through time, which would imply to observe a gradual decrease of $\delta_{\mathcal{V}}$.

Second, we estimated the gravity-related torques $\hat{\mathbf{G}}(q_{\mathrm{e}}, q_{\mathrm{s}}) = [\hat{\tau}_{\mathrm{e}}, \hat{\tau}_{\mathrm{s}}]'$ as a function of the participant's posture using common dynamics,

$$\hat{\mathbf{G}}(q_{\mathrm{e}}, q_{\mathrm{s}}) = \begin{bmatrix} \hat{\tau}_{\mathrm{s}} \\ \hat{\tau}_{\mathrm{e}} \end{bmatrix} = g \begin{bmatrix} (m_{\mathrm{a}} l_{\mathrm{a}} \alpha_{\mathrm{a}} + m_{\mathrm{f}} l_{\mathrm{a}}) \sin(q_{\mathrm{s}}) + m_{\mathrm{f}} l_{\mathrm{f}} \alpha_{\mathrm{f}} \sin(q_{\mathrm{s}} + q_{\mathrm{e}}) \\ m_{\mathrm{f}} l_{\mathrm{f}} \alpha_{\mathrm{f}} \sin(q_{\mathrm{s}} + q_{\mathrm{e}}) \end{bmatrix} \quad (3)$$

where $q_{\mathrm{e}}$ and $q_{\mathrm{s}}$ are the measured elbow and shoulder joint angles, $m_{\mathrm{a}}$ and $m_{\mathrm{f}}$ are the masses of the arm and forearm, $l_{\mathrm{a}}$ and $l_{\mathrm{f}}$ are the lengths of the arm and forearm, $\alpha_{\mathrm{a}}$ and $\alpha_{\mathrm{f}}$ are the relative positions of the center of mass of the arm and forearm, allowing to obtain $G_{\mathrm{a}}$ and $G_{\mathrm{f}}$ (see Fig. 1B), and $\hat{\tau}_{\mathrm{e}}$ and $\hat{\tau}_{\mathrm{s}}$ are the estimated elbow and shoulder gravity torques. The length and masses of the segments and the relative positions of center of masses were computed using anthropometric tables[78]. Then, given the vibratory landscape was defined based on the exoskeleton's end-effector position, we estimated the evolution of gravity-related efforts as a function of the exoskeleton's posture, thereby defining $\hat{\mathbf{G}}(q_{\mathrm{e}}^{\mathrm{r}}, q_{\mathrm{s}}^{\mathrm{r}})$ where $q_{\mathrm{e}}^{\mathrm{r}}$ and $q_{\mathrm{s}}^{\mathrm{r}}$ are the exoskeleton's elbow and shoulder joint angles. As usual in such settings, the human and exoskeleton segments were not aligned due to the passive joints included in the ergonomic interfaces, implying $q_{\mathrm{e}} \neq q_{\mathrm{e}}^{\mathrm{r}}$ and $q_{\mathrm{s}} \neq q_{\mathrm{s}}^{\mathrm{r}}$. Therefore, to obtain the relationships $q_{\mathrm{e}}^{\mathrm{r}}(q_{\mathrm{e}}, q_{\mathrm{s}})$ and $q_{\mathrm{s}}^{\mathrm{r}}(q_{\mathrm{e}}, q_{\mathrm{s}})$, we fitted a 2nd order polynomial mapping between the human hand and exoskeleton end-effector position. An illustration of the resulting field, defined using the norm of gravity-related efforts $\| \hat{\mathbf{G}} \|$ as a function of the exoskeleton end-effector position in the task space $(x^{\mathrm{r}}, z^{\mathrm{r}})$, is provided in Fig. 2A. Finally, one can define the distance between the optimal point to reach with respect to the gravity-related efforts and the measured exoskeleton's end-effector position as follows,

$$\delta_{\mathbf{G}} = \left\| \begin{bmatrix} x_{\mathbf{G}}^{*} \\ z_{\mathbf{G}}^{*} \end{bmatrix} - \begin{bmatrix} x^{\mathrm{r}} \\ z^{\mathrm{r}} \end{bmatrix} \right\|, \text{ where } \begin{bmatrix} x_{\mathbf{G}}^{*} \\ z_{\mathbf{G}}^{*} \end{bmatrix} = \underset{(x^{\mathrm{r}}, z^{\mathrm{r}})}{\arg\min} \| \hat{\mathbf{G}} \| \quad (4)$$

The computation of this distance at the end of each reaching movement performed by the participant will serve as the main metric to quantify their adaptation to the gravity landscape and analyze whether they adopted more ergonomic postures through time, which would imply observing a gradual decrease of $\delta_G$. Values reported in the present paper were obtained by averaging $\delta_V$ and $\delta_G$ over the last 3 s of each trial, consistently with the validation duration.

## Manifold reaching task

Participants were asked to perform blocks of reach-to-hold movements in a parasagittal plane using shoulder and elbow flexion/extension movements (see Fig. 1). The start posture of participants was $q_s = 0°$ and $q_e = 90°$, which was controlled in real-time using the motion capture system (see shaded posture in Fig. 1A). The movements were directed toward a 80 cm high and 5 cm large vertical bar displayed on a large screen, thereby defining a manifold in which any point was a valid target. Participants were placed at 80 cm from the screen, based on the **x**-axis distance between it and their shoulder. A 2.5 cm blue cursor allowed to show the current height and lateral position pointed by the exoskeleton's end-effector to participants as soon as they entered the target space. Simultaneously, a 3 s countdown was displayed on the screen. During the countdown, participants had to keep the cursor within the target to validate the trial. If the cursor went outside of the target, e.g., because of the vibration, then the trial failed. The target volume's threshold was set whenever $x^r > 62.5$ cm, with the $(\mathbf{x}, \mathbf{z})$ frame's origin placed at the level of the human shoulder. This allowed for ensuring that reaching the target zone required sufficient movements from the participants while allowing them to easily reach a wide range of vertical positions. Finally, the target volume's dimensions were $17.5 \times 5 \times 80$ cm, in the $(\mathbf{x}, \mathbf{y}, \mathbf{z})$ frame of Fig. 1A.

A trial was considered successful if the participant was able to hold their final reaching posture, with the end-effector pointing inside the vertical target, for the whole countdown. In that case, it meant that they were able to compensate for the vibrations. Conversely, a trial was considered failed before the end of the countdown if (i) the participant retracted their arm outside of the vibration zone, (ii) they were not able to maintain the end-effector at a constant height, or (iii) they let the vibrations push them outside of the vertical bar laterally. The order and requirements of the experimental blocks were as follows (see Fig. 1C),

- ET – ET in transparent mode: participants had to perform 10 reach-to-hold movements toward the ten predefined targets of height $z_i^E$, $i \in [[1, 10]]$ (one movement for each). The trials did not need to be successful.
- T – Transparent reaching: participants had to perform 40 successful reach-to-hold movements toward the vertical bar. The exoskeleton was only controlled in transparent mode, which means that participants were only subject to gravity when maintaining their final posture. This block was performed twice.
- E$V$ – ET of gravito-vibratory landscape: participants had to perform 10 reach-to-hold movements toward the predefined targets as in ET. The trials did not need to be successful.
- $V$ – reach-to-hold in gravito-vibratory landscape: participants had to perform 20 successful reach-to-hold movements toward the predefined targets. The exoskeleton was controlled in transparent mode outside of the vibration zone. This block was performed four times.
- AE – AE: participants had to perform 40 successful reach-to-hold movements toward the vertical bar. The exoskeleton was only controlled in transparent mode. This block was performed twice.

The ET blocks allowed participants to build a prior for each component of the gravito-vibratory landscape, with the objective of facilitating the motor ET process during the free reaching blocks. To avoid fatigue, breaks of at least 1 minute were taken between each block.

Importantly, $z^c$ was different for each participant, which allowed us to normalize the change in final posture height requested to minimize the vibrations. It was set either 20 cm above or 20 cm below the median final height of the participant during the transparent blocks. Two groups of participants were thereby constituted, the "above" and the "below" groups respectively. The *above* group was constituted of participants who selected final postures below their shoulder height on median, and conversely for the *below* group.

## Statistical analyses

We used linear mixed models to assert the main effects of the different conditions and groups on the main studied variables {DV} while accounting for all the trials performed by each participant. Note that we verified the linearity, homoscedasticity, normality and independence of the errors in our distributions to ensure the validity of the tests. These models were as follows,

$$\text{DV} \sim \text{cond} * \text{group} + 1|\text{S} \qquad (5)$$

where cond $\in \{T, V, AE\}$ is the tested condition and group $\in$ {downwards, upwards} is the group to which the participant S belongs. This model was then compared to another linear mixed model, without any fixed effect, using a likelihood ratio test, which indicated whether any significant effect of the "cond*group" term existed. The linear mixed models analyses were conducted using the *pymer*4 python package[79].

In case of a significant effect, post-hoc comparisons were performed using $t$-tests with a Tukey correction to verify the differences between fixed effects. The level of significance of all the performed tests was set at $p < 0.05$ and we reported the Cohen's $D$ as a measure of effect sizes. Whenever the Cohen's $D$ was below 0.4, the effect was flagged as small. The post-hoc analyses were performed using Jasp[80].

## Data availability

All data supporting the findings of this study are available within the paper.

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

## Acknowledgements

We thank Anaïs Farr and Myriam Marchesseau for their help during the Investigation. This study was funded by the French Agence Nationale de la Recherche (EXOMAN project, grant number ANR-19-CE33-0009).

## Author contributions

Conceptualization – W.G., B.B., N.V., D.V. Methodology – W.G., D.V. Software – W.G., L.Q., D.V. Validation – all authors. Formal analysis – W.G., L.Q. Investigation – W.G. Resources – B.B., N.V. Data Curation – W.G., L.Q. Visualization – W.G., L.Q., D.V. Supervision – B.B., N.V., D.V. Project administration – B.B., N.V. Funding acquisition – B.B., N.V. Writing - Original Draft – W.G., D.V. Writing - Review & Editing – all authors.

## Competing interests

The authors declare no competing interests.
