## [Transparent Peer Review file · Communications Engineering]

How task-relevant vibratory feedback from an active exoskeleton can lead to ergonomic postures

Corresponding Author: Dr Dorian Verdel

Version 0:

Reviewer comments:

Reviewer #1

(Remarks to the Author)

Main Claims

This manuscript proposes a novel approach to improve postural adaptation and learning through vibrotactile feedback during a task performed with an upper-limb exoskeleton. The key claim is that task-relevant perturbations, delivered through a vibratory field centered around a specific region of the workspace, can induce postural shifts in participants, thereby promoting adaptation towards biomechanically favorable zones. A second claim is that this approach avoids potential issues with cognitive overload or "neural resource allocation problems" that can arise from multiple sensory channels by embedding the perturbation within the motor task itself.

These claims are relevant and timely, as they address important open challenges in shared control, human-exoskeleton interaction, and motor learning. The strategy of using implicit cues (vibration) to shape behavior without overt instructions may have strong implications for both assistive and rehabilitative robotic systems.

Overall, the manuscript presents the following major strengths

- The study is well motivated, and the general idea of using task-relevant variability to guide learning is grounded in current neuroscience literature.
- The experimental setup and robotic system are carefully described.

The paper is clear in its structure and, overall, communicates the concept effectively.

However, some points need clarification or further justification. Also, the manuscript needs to be proofread. There are some sentences that are written in an awkward manner and would benefit from revision to improve the overall clarity.

1. Definition and Calculation of Posture Preference

It is not entirely clear how the preferred posture is determined. The reader is left to assume that, during the transparent block (T), the participant performs 40 trials to a self-selected position; however, this is not explicitly stated in the manuscript.

Additionally, the role of the 10 postures at 10 different heights performed during the ET block is unclear. Further clarification on these points would improve the manuscript's transparency and help readers better understand the experimental protocol.

1. Unclear Methodological Structure and Timeline

The explanation of the experimental blocks is fragmented and only clarified in the methods section after the results. In its current form, the manuscript feels like a puzzle, requiring the reader to piece together information to understand the study. Frequently, the reader is left making assumptions that are later contradicted by new information. It would benefit the reader to have a timeline or schematic earlier in the manuscript, or at least to summarize the structure and purpose of each block before presenting the findings. The fact that information is dispersed makes it very difficult for the reader to understand in depth the study.

2. Clarification of Spatial Feedback Activation (x_r threshold)

According to the methods, feedback is activated when $x_r > 62.5$ cm, yet the exploration described is primarily vertical. You mention that the amplitude is modulated by the distance between the center of the landscape and the exoskeleton end-effector. However, several details remain unclear. What is the distance between the participant and the vertical bar? You state that the vertical bar is 80 x 5 cm (height x width) and displayed on a screen, but what is the distance between the screen and the participant? When you refer to $x_r > 62.5$ cm, where exactly is the end-effector positioned relative to the target? How far from it? Additionally, what is the purpose of the countdown? Is it the time the participant needs to hold the position, or is the time allowed to correct their posture? More detail is needed to fully understand the experimental setup and ensure reproducibility. Also, an image of the real setup could help to better visualize it.

10. Ergonomic Framing Without Ergonomic Validation

After reading the manuscript, we found the title somewhat misleading. As we understand it, the study does not actually guide participants toward a more ergonomic position (in fact, some of the participants are led towards less ergonomic postures). Instead, the main focus appears to be on validating the biofeedback mechanism, specifically whether participants can follow and maintain the indicated posture. It is recommended to reconsider the title or address this issue more thoroughly in the manuscript, as the current title does not accurately reflect the study's content and objectives. Furthermore, there is no evidence presented that the center of the vibratory field (z_c) corresponds to a more ergonomic posture than $z_{\text{preferred}}$. Without such evidence, the claim of ergonomic benefit remains speculative. In fact, the manuscript presents no evidence that either the above or below postures offer superior ergonomic benefits.

Additionally, the definition of ergonomic posture in this study is based solely on the effort required for each position, without reference to any established ergonomic scale, model, or validation. Further, regarding equation 3, although the authors state that masses and lengths were calculated using anthropometric tables, there is no discussion of how Equation 3 generalizes across individuals with different anthropometric characteristics and the actual used values are not mentioned.

4. Neural Resource Allocation Problem

This concept is briefly referenced (lines 36–37 and 53) but remains vague. Particularly given the context of arm movement, where a haptic motor could be placed on the upper arm to guide motion. Is the underlying concern cognitive load, sensorimotor competition, or another mechanism? Providing clarification and explicitly connecting this point to reference [27] would strengthen the theoretical basis of the manuscript.

It would be interesting to compare the differences in cognitive load and intuitiveness between using a standard haptic motor and your current method to better validate your assumptions. Furthermore, if the task involved bimanual movements, would it require two separate haptic signals?

5. Task-Relevant Variability and Perturbation Definition

The manuscript emphasizes task-relevant variability, but it is not entirely clear how the vibration acts as a task-relevant cue. Until the results section the reader is left without clear idea how haptic feedback is provided to the participant. Whether it is during the full movement or if it is only at the final position. Therefore, when on line 60 we read “participants were exposed to haptic stimulation, implemented as horizontal vibrations applied by the robot in certain conditions, pushing them outside of the target so as to be task-relevant” it is confusing how subjecting the participant to forces that push in a direction that is different from the goal of the task could be task relevant.

After reading figure 1 and the results, the reader is able to understand that vibration seems to be only applied once the participants reach their final goal, and that possibly that is why the authors meant by task relevant. However, this information needs to be provided earlier in the manuscript as it is confusing and misleading.

Additionally, regarding the mechanism used to provide haptic feedback, the idea of “pushing the participant outside of the target” seems somewhat paradoxical if the vibration is active near the desired goal. Is the feedback mechanism designed to penalize being out of the correct position, rather than to reinforce proximity to the target? Although it was stated on the methods that amplitude was modeled according to the distance to the target point, more detail on the haptic field is structured and how task relevance is ensured would be helpful.

6. Confounding Role of Natural Exploration

During the T phase, participants perform 40 reach-to-hold movements. Although these movements are directed to targets, participants are free to stabilize posture within a spatial zone. Thus, they may naturally discover a biomechanically comfortable or efficient position without feedback. The observed adaptations might stem from this exploration rather than the vibration per se. Although the AE phase attempts to isolate learning effects, the data does not eliminate this alternative explanation.

8. Statistical Robustness and Power

The results shown (e.g., in Figure 5) suggest trends in adaptation, but the manuscript lacks discussion on statistical power. Are the sample sizes sufficient to support the conclusions? Were power analyses conducted in advance? This would be particularly relevant for the interpretation of AE (after-effect) phases.

9. Reversion to Preferred Posture in AE Phase

Figure 5 suggests that once the vibratory stimulus is removed, participants tend to revert back toward their original preferred posture. This raises the question of whether vibratory stimulation leads to durable motor learning or merely a transient shift. This point could be better discussed.

Further, was the waiting period between the V blocks and the AE blocks also 1s? This is a short period. Could you discuss whether the observed effects would persist over a longer interval, and whether additional repetitions or an alternative training structure might be necessary to ensure that the results are both generalizable and sustained over time?

10. Other concerns

The authors state that they used Motion capture data, but its purpose is not clear.

At line 60 the authors mentioned horizontal vibrations, while in line 70 it was mentioned “with a lateral vibration applied by the exoskeleton”. Please maintain consistency during the manuscript as it eases interpretation.

It appears that participants were required to guide the exoskeleton's end-effector to the target point, rather than using their own hand. Considering that the aim is to develop solutions for industrial applications and promote more ergonomic postures, I question whether this strategy is optimal. Guiding the end-effector may result in larger shoulder or elbow angles, potentially compromising ergonomics. Would it not make more sense to focus on teaching participants to guide their own hand, rather than the exoskeleton's end-effector position?

In figure 2 you state “Landscapes are computed from 0.4m below the shoulder to 0.1m above the shoulder”. Were these values chosen because of the exoskeleton dimensions? Please justify these values.

Impact and Relevance

The ideas explored in the paper are likely to stimulate further research in the area of implicit guidance in human-robot

interaction. The task-relevant vibration field introduces a novel method for promoting learning without explicit instruction, which could be beneficial in rehabilitation and assistive contexts. However, the durability of the learning, the interpretability of the vibratory feedback, and the grounding of the "preferred posture" construct require further substantiation. Further, the paper is dense. I took several

Reviewer #2

(Remarks to the Author)

I co-reviewed this manuscript with one of the reviewers who provided the listed reports. This is part of the Communications Engineering initiative to facilitate training in peer review and to provide appropriate recognition for Early Career Researchers who co-review manuscripts.

Reviewer #3

(Remarks to the Author)

In this paper, the authors proposed to use exoskeletons with vibrations as biofeedback to guide users toward ergonomic postures. I found the motivation interesting and the experiments were well designed and analyzed. And it was also interesting to see from the results that people would be able to adapt towards the ergonomic postures with the exoskeleton and vibrations.

I do have the following questions or comments.

1. I think for the reaching-and-hold task, the end position is usually important and fixed. From the current study design, it seemed that you allowed the participants to explore different postures while being able to adjust the end-effector position as well. From what I stand, what we really want to guide people towards is their intermittent joint position, for example, they can have a larger elbow angle and lower shoulder angle in order to maintain the same final position but with lower overall gravity landscape. If the final position is changeable, what is the point of having an ergonomic posture? I think authors should make that more clear in the paper.

2. I think authors should bring to the very front why there are above and below groups in the experiment. It was very confusing until late in the results to explain the reason.

3. Fig 1C: Does the dashed line represent the final posture of the exoskeleton for different trial? Or it just connect the start and end position of the exoskeleton? I do have trouble understanding the plots for a long time and I think it could be made more clear for easier understanding.

Version 1:

Reviewer comments:

Reviewer #1

(Remarks to the Author)

The authors have clearly made an effort to improve the manuscript in response to earlier feedback. The revised version is more transparent and easier to follow, and it reflects meaningful progress in clarifying the contributions. That said, there remain some issues that need to be addressed before the manuscript can be considered for publication. These issues concern both methodological clarity and writing quality, which are required to make the manuscript suitable for publication.

Our biggest concern at this stage is that there is no explicit statement in the statistical analysis section confirming that the assumptions of linear mixed models were verified. Specifically, the authors should make a statement if they checked:

- The explanatory variables are linearly related to the response,
- The errors have constant variance (homoscedasticity),
- The errors are independent, and
- The errors are normally distributed.

Without this information, the validity of the statistical conclusions remains uncertain.

In addition, we believe a general revision of the writing is still necessary. The manuscript remains dense in places, and some sentences need to be re-read to be fully understood. Connectors are sometimes overused, which makes the text unnecessarily heavy. For example, the word "importantly" appears multiple times in close succession. A careful stylistic revision would improve the manuscript's flow.

Research questions (Lines 82–83): In my view, RQ2 and RQ3 could be merged into a single, clearer question: "Do participants retain learning from the exoskeleton?" The two-group design is not a separate research question but rather a methodological approach to address this issue, clarifying whether the observed effects result from adopting a naturally less effortful posture or from actual retention of learning under a less natural condition. Presenting them as distinct RQs therefore adds unnecessary complexity without improving clarity.

Minor comments:

Figure 1C caption: The authors refer to "top" and "bottom," but the figure is displayed horizontally. Using "left" and "right" would be clearer and avoid confusion.

Ergonomic framing: Although the addition of ergonomics was requested, the authors now repeat the ergonomic assumptions three times, which makes the text repetitive for the reader.

Reviewer #2

(Remarks to the Author)

I co-reviewed this manuscript with one of the reviewers who provided the listed reports. This is part of the Communications Engineering initiative to facilitate training in peer review and to provide appropriate recognition for Early Career Researchers who co-review manuscripts.

Reviewer #3

(Remarks to the Author)

I think the revised manuscript adequately addressed all my previous questions and comments. I have no additional comments.

Version 2:

Reviewer comments:

Reviewer #1

(Remarks to the Author)

I believe the revised manuscript has satisfactorily addressed all our previous questions and remarks. We have no further comments.

Reviewer #2

(Remarks to the Author)

I co-reviewed this manuscript with one of the reviewers who provided the listed reports. This is part of the Communications Engineering initiative to facilitate training in peer review and to provide appropriate recognition for Early Career Researchers who co-review manuscripts.

Answer to reviewers:

We would like to thank the reviewers and editors for their constructive comments that contributed to greatly improve the manuscript. We provide below a point-to-point response to the reviewers' comments. Our answers appear in blue and the modifications in the revised manuscript are also highlighted in blue for the sake of clarity.

1 Reviewers #1 and #2

General assessment: This manuscript proposes a novel approach to improve postural adaptation and learning through vibrotactile feedback during a task performed with an upper-limb exoskeleton. The key claim is that task-relevant perturbations, delivered through a vibratory field centered around a specific region of the workspace, can induce postural shifts in participants, thereby promoting adaptation towards biomechanically favorable zones. A second claim is that this approach avoids potential issues with cognitive overload or "neural resource allocation problems" that can arise from multiple sensory channels by embedding the perturbation within the motor task itself.

These claims are relevant and timely, as they address important open challenges in shared control, human-exoskeleton interaction, and motor learning. The strategy of using implicit cues (vibration) to shape behavior without overt instructions may have strong implications for both assistive and rehabilitative robotic systems. Overall, the manuscript presents the following major strengths:

- The study is well motivated, and the general idea of using task-relevant variability to guide learning is grounded in current neuroscience literature.
- The experimental setup and robotic system are carefully described.

The paper is clear in its structure and, overall, communicates the concept effectively.

However, some points need clarification or further justification. Also, the manuscript needs to be proofread. There are some sentences that are written in an awkward manner and would benefit from revision to improve the overall clarity.

We thank the reviewers for their in depth assessment of our manuscript. We provide a point-by-point answer to the reviewers' concerns and comments below.

1.1 Definition and Calculation of Posture Preference

Comment 1: It is not entirely clear how the preferred posture is determined. The reader is left to assume that, during the transparent block (T), the participant performs 40 trials to a self-selected position; however, this is not explicitly stated in the manuscript.

We agree that this point is important and was not made clear in the manuscript. Indeed, the preferred height of the participants was defined as the median height reached across the 40 initial trials in transparent mode. To clarify this point, we have included the following statements in the manuscript:

- Caption of Fig. 1 (p.3): "*This preferred endpoint height is defined as the median height reached across 40 trials, performed during the transparent block (T).*"
- First paragraph of results (p.4, l.93): "*[...], by extracting the median reached height of the 40 trials performed in this mode*"

Comment 2: Additionally, the role of the 10 postures at 10 different heights performed during the ET block is unclear. Further clarification on these points would improve the manuscript's transparency and help readers better understand the experimental protocol.

The role of the ET exploration was indeed not specified in the manuscript. This block was used to allow the participants to build an internal model of the efforts and comfort associated with any height within the target space, as well as to remain consistent with the exploration phase provided with the vibration activated. To clarify these points, we have clarified Fig. 1B and added the following statement at the beginning of the Results (p.4, l.88):

“During the ET block, participants reached and held 10 predefined postures equally spread across the workspace, which (i) allowed them to build a prior of the efforts and comfort associated with postures across the target space, and (ii) allowed consistency with the \mathcal{V} conditions that are preceded by the same exploration to build a prior regarding the task with added task-relevant variability. Then, the transparent blocks [...]”

1.2 Unclear Methodological Structure and Timeline

Comment 3: The explanation of the experimental blocks is fragmented and only clarified in the methods section after the results. In its current form, the manuscript feels like a puzzle, requiring the reader to piece together information to understand the study. Frequently, the reader is left making assumptions that are later contradicted by new information. It would benefit the reader to have a timeline or schematic earlier in the manuscript, or at least to summarize the structure and purpose of each block before presenting the findings. The fact that information is dispersed makes it very difficult for the reader to understand in depth the study.

From this comment, we understand that our Fig. 1B and associated caption were not clear enough, as they contained the required information. Consequently, we modified the structure of Fig. 1 (p.3) so as to make clear that the B panel presents the order of the blocks. The rationale for the different blocks is now presented in the first paragraph of the Results section (see answers above). We also added the following sentence at the beginning of the panel’s description (p.3): *“This panel presents the structure of the experiment.”*

1.3 Clarification of Spatial Feedback Activation (x^r threshold)

Comment 4: According to the methods, feedback is activated when $x^r > 62.5$ cm, yet the exploration described is primarily vertical.

Based on the reviewers’ comment, we think that this aspect of the methods was not clear enough. In our experiment, participants were asked to reach towards the vertical bar displayed on a screen and hold their final posture. In all the conditions, and especially those where the vibrations were applied, we wanted to make sure that the arm of the participant was sufficiently extended. Therefore, we set a threshold based on the horizontal distance (i.e. on the x axis) between the shoulder of the participant and the end-effector of the exoskeleton. When this threshold was crossed, the target turned green and the vibration started in the conditions where it was present. To clarify this point, we added the following statement in the caption of Fig. 2:

“Importantly, this interval on the x axis represents the range of valid reaching depths, in which (i) the height-dependent vibration was activated in the EV and \mathcal{V} blocks, and (ii) the task could be validated.”

Comment 5: You mention that the amplitude is modulated by the distance between the center of the landscape and the exoskeleton end-effector. However, several details remain unclear. What is the distance between the participant and the vertical bar? You state that the vertical bar is 80 x 5 cm (height x width) and displayed on a screen, but what is the distance between the screen and the participant? When you refer to $x^r > 62.5$ cm, where exactly is the end-effector positioned relative to the target? How far from it?

We have now clarified those points. In particular, we now clearly define the distance between the participant’s shoulder and the screen/target (i.e. 80 cm), and the depth of the reachable space (i.e. 17.5 cm in front of the bar) allowing participants to adapt how far they reach. These settings allowed all the participants to easily reach the target space and adapt their final posture (on both the x and z axes) within this target space. These details have been added in the Manifold reaching task subsection of the Methods:

- p.11, l.383: *“Participants were placed at 80 cm from the screen, based on the x -axis distance between it and their shoulder.”*

- p.12, l.390: “Finally, the target volume’s dimensions were 17.5 cm × 5 cm × 80 cm, in the (x, y, z) frame of Fig. 1A.”

Comment 6: Additionally, what is the purpose of the countdown? Is it the time the participant needs to hold the position, or is the time allowed to correct their posture? More detail is needed to fully understand the experimental setup and ensure reproducibility. Also, an image of the real setup could help to better visualize it.

We agree with the reviewer that this information was missing. During the timer, participants had to keep the cursor within the target, otherwise the trial was failed. The following statement has been added in the Manifold reaching task subsection of the Methods to clarify this point (p.11, l.386):

“During the countdown, participants had to keep the cursor within the target to validate the trial. If the cursor went outside of the target, e.g. because of the vibration, then the trial was failed.”

1.4 Ergonomic Framing Without Ergonomic Validation

Comment 7: After reading the manuscript, we found the title somewhat misleading. As we understand it, the study does not actually guide participants toward a more ergonomic position (in fact, some of the participants are led towards less ergonomic postures). Instead, the main focus appears to be on validating the biofeedback mechanism, specifically whether participants can follow and maintain the indicated posture. It is recommended to reconsider the title or address this issue more thoroughly in the manuscript, as the current title does not accurately reflect the study’s content and objectives.

We thank the reviewer for this comment as it allowed us to clarify the hypotheses that are investigated in our work. We have now included several statements clarifying that (i) the work is indeed focused on validating the feedback mechanism in different contexts, and (ii) actually allows to improve posture after the interaction, even in case the implicit guidance was towards a worse posture, hence the title. Nevertheless, we agree that the previous title was not specific enough, therefore we modified it to “**How task-relevant vibratory feedback from an active exoskeleton can lead to ergonomic postures**”. The statements in the Introduction are as follows:

- p.2, l.63: “Therefore, in the present paper, we focus on the general feasibility of the method, without selecting targeted postures based on a specific ergonomic score. To remain as general as possible, we investigate two cases: one where the targeted posture is better and one where it is worse than the participant’s preferred one.”
- New statements in bold, p.2, l.80: “The investigations focus on **verifying three hypotheses: (i) whether participants succeed or fail to adapt by adopting a posture minimizing the intensity of vibrations and/or of gravity-related efforts, (ii) whether they can retain the adapted posture when it is better than their preferred one, which would be critical for ergonomics improvement (see Fig. 1C), and (iii) whether forcing the exploration of the target space, even towards poor postures, can still be beneficial as it could allow the user to build a better internal model of the task at hand.**”

Comment 8: Furthermore, there is no evidence presented that the center of the vibratory field (z_c) corresponds to a more ergonomic posture than $z_{preferred}$. Without such evidence, the claim of ergonomic benefit remains speculative. In fact, the manuscript presents no evidence that either the above or below postures offer superior ergonomic benefits.

We thank the reviewer for this comment as it allowed us to improve our description of the two postures towards which participants were implicitly guided. Specifically, we give a better description of the ergonomics criteria that can be used to discriminate between the *above* and *below* postures throughout the paper:

- New statements in bold, Introduction, p.2, l.73: “When the minimum of the vibrations was above the preferred height, **three ergonomic features were impacted: (i) gravity-related efforts were increased (see Fig. 2A); (ii) shoulder and elbow angles were closer to their upper-limits [1, 2]; and (iii) blood-flow was reduced due to gravity and compression of the tendons in the shoulder, especially when z_c was above the shoulder [3, 4]. Conversely, when the minimum of the vibrations was below the preferred height, it mechanically led to an improvement of these three ergonomic criteria.**”

- Results p.4, l.107: *“This separation in two groups allows to test our method’s ability to implicitly guide users towards a specific posture in two contrasted contexts. First, in the case of the below group, the vibration cancels at a more ergonomic final posture than the participant’s preferred one with (i) reduced gravity-related effort, (ii) further from the shoulder and elbow joints upper-limits, and (iii) no reduced blood-flow due to gravity. Therefore, the below group allows to test the first two hypotheses presented at the end of the Introduction. Conversely, for the above group, these three criteria are made worse when canceling the vibration. Therefore, the above group allows to test the first and third hypotheses presented at the end of the Introduction.”*
- Discussion (p.9, l.253): *“Importantly, in the present paper a posture was considered more ergonomic if it was lower in the target space as – beyond gravity-related torques analyzed here that are the major biomechanical factor in the task that may lead to musculoskeletal disorders [4] – it leads to joint angles farther from elbow and shoulder upper-limits [2], which is related to better blood-flow in the upper-limb [3].”*

Comment 9: Additionally, the definition of ergonomic posture in this study is based solely on the effort required for each position, without reference to any established ergonomic scale, model, or validation.

Despite our focus on effort, please note that we actually cite relevant papers regarding established ergonomic scales in the Discussion, whether it be purely posture-based [5, 6] or also including an estimation of effort [2]. However, we agree these considerations were not properly addressed when discussing the postures at which the vibration canceled. We believe it is now clearer (see comment 8).

Comment 10: Further, regarding equation 3, although the authors state that masses and lengths were calculated using anthropometric tables, there is no discussion of how Equation 3 generalizes across individuals with different anthropometric characteristics and the actual used values are not mentioned.

We agree with the reviewer that anthropometric tables are not a perfect tool to estimate human limbs dynamics characteristics, as they can be inaccurate for users with different body morphologies. However, they are still a valuable tool to show trends and compare the effects of the exoskeleton between conditions. To clarify how this inaccuracy could be solved in practice, we have added the following statement in the Discussion (p.9, l.277):

“It would also be interesting to evaluate the benefits of the method with more accurate techniques than anthropometric tables, for example by identifying the human limbs dynamics from interaction efforts [7, 8].”

1.5 Neural Resource Allocation Problem

Comment 11: This concept is briefly referenced (lines 36–37 and 53) but remains vague. Particularly given the context of arm movement, where a haptic motor could be placed on the upper arm to guide motion. Is the underlying concern cognitive load, sensorimotor competition, or another mechanism? Providing clarification and explicitly connecting this point to reference [27] would strengthen the theoretical basis of the manuscript.

We thank the reviewer for their comment as our previous description was indeed too limited for a clear understanding by the reader. We now better explain this concept and why it is relevant to account for it in our context. To make this clearer, we added the following statement in the Introduction (p.2, l.35):

“This concept was recently introduced to discuss the limited resources available to the brain to simultaneously control the movements of natural limbs and supernumerary limbs (SLs), which provide additional degrees-of-freedom to the human body. This control requires both to process additional sensory signals associated with SLs and generate relevant motor commands for SLs. In this context, the finiteness of human neural resources may lead to decreased natural limbs control performance when interacting with SLs. This concept also applies to biofeedback methods, as they generate additional sensory stimulation to those normally processed by the CNS for a given task. Such stimulation create a risk of diverting neural resources from their usual role, inducing an increased cognitive cost to maintain performance.”

Comment 12: It would be interesting to compare the differences in cognitive load and intuitiveness between using a standard haptic motor and your current method to better validate your assumptions. Furthermore, if the task involved bimanual movements, would it require two separate haptic signals?

First, we agree with the reviewer that the next step will be to compare our method to standard vibrotactile biofeedback and to other methods such as auditive feedback. We now acknowledge this point with the following statement in the Discussion (p.9, l.280):

“Although systematic comparisons between our method and traditional biofeedback techniques will be needed to assess differences in performance and resulting user benefits, several functional findings can already be made.”

Second, in the case of a bimanual task, the haptic signal would not necessarily need to be differentiated. We have added the following statement in the Discussion to clarify this point (p.9, l.294):

“Although, in the present paper, we have only investigated the case of a unimanual task, we argue that bimanual tasks could also be treated using our method, whenever the user can be equipped with a bilateral active exoskeleton. In practice, the implementation would depend on whether the two hands are mechanically coupled, e.g. when carrying a large box, or not, e.g. when holding one object in each hand. In the first case, the haptic perturbation could be common, depending on the pose of the manipulated object, whereas in the other case, signals provided on each side can be independent.”

1.6 Task-Relevant Variability and Perturbation Definition

Comment 13: The manuscript emphasizes task-relevant variability, but it is not entirely clear how the vibration acts as a task-relevant cue. Until the results section the reader is left without clear idea how haptic feedback is provided to the participant. Whether it is during the full movement or if it is only at the final position. Therefore, when on line 60 we read “participants were exposed to haptic stimulation, implemented as horizontal vibrations applied by the robot in certain conditions, pushing them outside of the target so as to be task-relevant” it is confusing how subjecting the participant to forces that push in a direction that is different from the goal of the task could be task relevant.

We have now clarified these points. Importantly, when we refer to task-relevant variability, we mean that it directly impacts the successful completion of the task, thereby prompting the CNS to adapt. This notion of task relevance is based on previous works in movement neuroscience, that showed that variability not impacting the task completion (i.e. task-irrelevant) may not be corrected by the brain during movement control [9]. The following statements (in bold) were added in the Introduction:

- p.2, l.49: *“Instead, we propose to use the exoskeleton to introduce task-relevant variability **via a haptic signal**, with the aim to promote the adaptation of the central nervous system (CNS) by generating motor errors. **In other words, if the human does not adapt to the feedback provided by the exoskeleton, they will not be able to do the task.** This task-relevant characteristic is critical as it is also known that task-irrelevant variability, **i.e. not hampering successful task completion**, may be ignored by the human sensorimotor system [9], which could prevent any significant adaptation.”*
- p.2, l.70: *“(iii) participants were exposed to a haptic stimulation, implemented as lateral vibrations applied by the robot **at the end of the reaching phase** to push them outside of the target, **thereby ensuring its task-relevance (vertical vibrations would not prevent the successful completion of the task)**”*

Comment 14: After reading figure 1 and the results, the reader is able to understand that vibration seems to be only applied once the participants reach their final goal, and that possibly that is why the authors meant by task relevant. However, this information needs to be provided earlier in the manuscript as it is confusing and misleading.

We agree with the reviewers’ comment. This information is now presented in the description of the task in the Introduction (see answer to comment 13).

Comment 15: Additionally, regarding the mechanism used to provide haptic feedback, the idea of “pushing the participant outside of the target” seems somewhat paradoxical if the vibration is active near the desired goal. Is the feedback mechanism designed to penalize being out of the correct position, rather than to reinforce proximity to the target? Although it was stated on the methods that amplitude was modeled according to the distance to the target point, more detail on the haptic field is structured and how task relevance is ensured would be helpful.

There might have been a confusion between the “target space” and the center of the vibration landscape “ z_c ” in this comment. The vibration is indeed lateral, with an intensity increasing based on the vertical distance between z_c and the

reached height. However, this does not mean that z_c is the target, although it could be considered a desired posture. In fact, the target space is unchanged between the initial blocks and the blocks with the vibration. Furthermore, we do not provide any visual feedback to participants regarding the position of z_c . The rationale is to investigate whether participants find z_c on their own, thereby choosing a posture consistent with the minimization of task-relevant variability whether it be better or worst with respect to other ergonomic factors. We have clarified these points through (i) the statements answering comment 13, and (ii) the following statements:

- Introduction, p.1, l.21: “*In the present study, we investigate a method using active exoskeletons to implicitly guide users towards ergonomic postures in tasks where multiple postures may be used, as it is usually the case in daily-life.*”
- Results, p.4, l.97: “*The assumption behind our method being that participants can be implicitly guided towards a specific posture with task-relevant haptic perturbations, no visual feedback is provided to the participants regarding the position of z_c in the target space.*”

1.7 Confounding Role of Natural Exploration

Comment 16: During the T phase, participants perform 40 reach-to-hold movements. Although these movements are directed to targets, participants are free to stabilize posture within a spatial zone. Thus, they may naturally discover a biomechanically comfortable or efficient position without feedback. The observed adaptations might stem from this exploration rather than the vibration per se. Although the AE phase attempts to isolate learning effects, the data does not eliminate this alternative explanation.

We understand the reviewer’s comment and can agree with it to some extent. Specifically, we agree that in the absence of a control group performing the same number of movements but always in transparent mode, one cannot definitely guarantee that natural movement variability could not lead to the same adaptation trends as those observed during and after exposure to our haptic feedback. However, this is very highly unlikely for several reasons. First, the change in posture is brutal during $\mathcal{V}1$ whereas it remained stable around their preferred posture in T for 40 movements, with significant differences compared to T that are in the same direction as implicitly transmitted by the vibration. This is for instance illustrated by Fig. 3. Second, participants refine their posture optimization through blocks, as shown by the decreasing trend in δ_V . Third, participants implicitly guided towards a more ergonomic posture (*below* group) tend to retain it, although we agree further studies are needed regarding long-term retention (see answer to comment 18). As we mention in the discussion, this does not imply that implicit exploration due to movement variability is not present. However, given the data, we believe it is very hard to argue that without the force field, this exploration through natural variability would lead to a similar result, including different trends between groups. Finally, the preliminary exploration phase was also introduced to minimize this confound by providing the users with prior knowledge about different arm postures.

1.8 Statistical Robustness and Power

Comment 17: The results shown (e.g., in Figure 5) suggest trends in adaptation, but the manuscript lacks discussion on statistical power. Are the sample sizes sufficient to support the conclusions? Were power analyses conducted in advance? This would be particularly relevant for the interpretation of AE (after-effect) phases.

We did not conduct a power analysis prior to performing the experiment. This is because such analyses require to have prior knowledge and plausible expectations regarding the distribution of results, which we did not have prior to the present experiment. Furthermore, it should be noted that such experiments can be quite difficult to carry out without putting too much strain on the exoskeleton as such systems can be quite fragile, which explains the relatively small sample size. Finally, although we agree it is not equivalent to a power analysis, we provide the effect size of all the statistical tests reported in the paper. In most cases, the effects are clear and large (Cohen’s $D > 0.8$) suggesting real differences between conditions and groups. In future studies over several experimental sessions, we will be able to investigate in depth whether the adaptations are retained over long periods, and to confirm the results presented in the present paper.

1.9 Reversion to Preferred Posture in AE Phase

Comment 18: Figure 5 suggests that once the vibratory stimulus is removed, participants tend to revert back toward their original preferred posture. This raises the question of whether vibratory stimulation leads to durable motor learning or merely a transient shift. This point could be better discussed. Further, was the waiting period between the V blocks and the AE blocks also 1s? This is a short period. Could you discuss whether the observed effects would persist over a longer interval, and whether additional repetitions or an alternative training structure might be necessary to ensure that the results are both generalizable and sustained over time?

We agree with the reviewers that this point was not enough discussed in the previous version of the manuscript. Consequently, we have added the following statement in the Discussion (p.9, l.273):

“Those studies, potentially involving repeated exposure to the gravito-vibratory landscape over several sessions, would allow to investigate whether the posture adaptations that we report can be sustained over long periods or, conversely, to analyze the time before they vanish post-exposure.”

As mentioned in the Methods, delays between blocks were at least one minute depending on the participants' felt fatigue. Usually, participants took slightly longer periods after having performed several blocks with the vibration activated. We did not measure the time taken by each participant, and only ensured that they all had at least one minute.

1.10 Other concerns

Comment 19: The authors state that they used Motion capture data, but its purpose is not clear.

Motion capture was mainly used to extract the real position of the exoskeleton's end-effector in the world frame. This could not be done by simply using the geometric model of the robot as its cable-based transmission is quite flexible, which generates effort dependent errors in pose estimation. Note that, in practice, this flexibility can be identified for a given robot, removing the need for an external measurement device. As was already mentioned in the manuscript, it was also used for post-processing to identify individualized mappings between the exoskeleton and the human's joints angles, allowing to estimate gravity-related efforts as a function of the robot's joints angles. Similarly to geometric errors, this mapping could be identified using the exoskeleton's embedded force/torque sensors, removing the need for an external device. This is now made clear in the Methods with the following statement (added elements in bold, p.10, l.326):

“Markers were also used to estimate the position of the exoskeleton's end-effector in the task space, thereby removing pose estimation errors due to the robot flexibility. In practice, both this identification and estimation can be performed using other information readily available in the exoskeleton, such as interaction forces [8].”

Comment 20: At line 60 the authors mentioned horizontal vibrations, while in line 70 it was mentioned “with a lateral vibration applied by the exoskeleton”. Please maintain consistency during the manuscript as it eases interpretation.

We have now corrected the terminology throughout the paper to consistently use the term “lateral”, whether it be to discuss the vibration or its effects on hand movements.

Comment 21: It appears that participants were required to guide the exoskeleton's end-effector to the target point, rather than using their own hand. Considering that the aim is to develop solutions for industrial applications and promote more ergonomic postures, I question whether this strategy is optimal. Guiding the end-effector may result in larger shoulder or elbow angles, potentially compromising ergonomics. Would it not make more sense to focus on teaching participants to guide their own hand, rather than the exoskeleton's end-effector position?

We understand the reviewers' comment. However, with our settings and exoskeleton, pretests showed that it was as intuitive for participants to use their hand or the exoskeleton's end-effector, but the latter led to more repeatable vibration application and a more rigorous protocol. This is because using markers directly placed on the hand of the participant can lead to inaccurate landscape computation due to finger and wrist movements. In practice, we agree that using a human limb reference to compute the landscape may be better. Such an improvement, that would also allow to compare movement with and without the exoskeleton, is feasible given a reliable geometric model of the robot and

a reliable mapping to compensate joints misalignments, which we did not have in the present study. Importantly, the misalignments are not very large when the arm is extended, so the projection of the hand and of the exoskeleton's end-effector on the screen are close.

Comment 22: In figure 2 you state "Landscapes are computed from 0.4m below the shoulder to 0.1m above the shoulder". Were these values chosen because of the exoskeleton dimensions? Please justify these values.

We have now clarified that this is just an illustration to highlight the evolution of the landscapes in the portion of the target space mostly reached by participants. In practice, the vibration was computed for any reached height.

1.11 Impact and Relevance

The ideas explored in the paper are likely to stimulate further research in the area of implicit guidance in human-robot interaction. The task-relevant vibration field introduces a novel method for promoting learning without explicit instruction, which could be beneficial in rehabilitation and assistive contexts. However, the durability of the learning, the interpretability of the vibratory feedback, and the grounding of the "preferred posture" construct require further substantiation. Further, the paper is dense. I took several [sentence cut in the received e-mail]

We thank the reviewer for their work that significantly contributed to improve the manuscript. Note that we have performed an overall scan of the manuscript to improve the writing quality and clarity.

2 Reviewer #2

I co-reviewed this manuscript with one of the reviewers who provided the listed reports. This is part of the Communications Engineering initiative to facilitate training in peer review and to provide appropriate recognition for Early Career Researchers who co-review manuscripts.

We thank the reviewer for their contribution to evaluating our work. The answers to the points raised by their review are provided above.

3 Reviewer #3

General assessment: In this paper, the authors proposed to use exoskeletons with vibrations as biofeedback to guide users toward ergonomic postures. I found the motivation interesting and the experiments were well designed and analyzed. And it was also interesting to see from the results that people would be able to adapt towards the ergonomic postures with the exoskeleton and vibrations.

We thank the reviewer for their assessment of our work and for their constructive feedback.

Comment 1: I think for the reaching-and-hold task, the end position is usually important and fixed. From the current study design, it seemed that you allowed the participants to explore different postures while being able to adjust the end-effector position as well. From what I understood, what we really want to guide people towards is their intermittent joint position, for example, they can have a larger elbow angle and lower shoulder angle in order to maintain the same final position but with lower overall gravity landscape. If the final position is changeable, what is the point of having an ergonomic posture? I think authors should make that more clear in the paper.

As mentioned in the manuscript, daily- and work-tasks performed by humans can usually be performed using an infinite number of solutions due to the redundancy allowed by human joints. Some of these solutions are ergonomic and some detrimental to the human. Here, we implemented a task aiming to reproduce these characteristics: the bar allows for target redundancy (and thus an infinity of valid arm postures for the task) and the hold component allows the effort cost of the task to be significant when repeated multiple times. Note that, in the present task, it is not possible to reach the same pose of the hand ($\mathbf{x}^h, \mathbf{z}^h$) in a parasagittal plane with several combinations of elbow and shoulder flexion/extension angles (q_s, q_e). Therefore, there is a strict equivalence between joint angles and hand position in the task (i.e. the function from joints to hand position is bijective). Then, the goal of the paper is to show that, when

multiple solutions are available, an implicit guidance based on task-relevant variability can lead the user to select a specific posture, e.g. chosen based on its RULA score. In the present paper, we tested the feasibility of the method and verified three hypotheses (see comment 2 below). This showed the method is feasible and beneficial in general, for both better and worse specified desired posture (i.e. implicit guidance is effective), without restricting to a specific definition of what is ergonomic. To clarify this point, we added the following statement in the Introduction (p.2, l.63):

“Therefore, in the present paper, we focus on the general feasibility of the method, without selecting targeted postures based on a specific ergonomic score. To remain as general as possible, we investigate two cases: one where the targeted posture is better and one where it is worse than the participant’s preferred one.”

We also completed the following sentence (in bold) in the Introduction (p.1, l.20):

*“In the present study, we investigate a method using active exoskeletons to guide users towards adopting and retaining ergonomic postures **in tasks where multiple postures may be used, as it is usually the case in daily life.**”*

Comment 2: I think authors should bring to the very front why there are above and below groups in the experiment. It was very confusing until late in the results to explain the reason.

We thank the reviewer for this comment that helped us better formulate the hypotheses that were investigated in the manuscript, thereby clarifying the rationale behind our protocol and the above/below groups. First, we have added the following information (in bold) at the end of the Introduction (p.2, l.80):

*“The investigations focus on **verifying three hypotheses: (i) whether participants succeed or fail to adapt by adopting a posture minimizing the intensity of vibrations and/or of gravity-related efforts, (ii) whether they can retain the adapted posture when it is better than their nominally preferred one, which would be critical for ergonomics improvement (see Fig. 1C), and (iii) whether the adaptation to the vibrations applied by the exoskeleton is robust, i.e. can also happen when the targeted posture is more effortful than the participant’s preferred posture.**”*

Second, we have added the following statement at the end of the first paragraph of the Results (p.4, l.107):

“This separation in two groups allows to test our method’s ability to implicitly guide users towards a specific posture in two contrasted contexts. First, in the case of the below group, the vibration cancels at a more ergonomic final posture than the participant’s preferred one with (i) reduced gravity-related effort, (ii) further from the shoulder and elbow joints upper-limits, and (iii) no reduced blood-flow due to gravity. Therefore, the below group allows to test the first two hypotheses presented at the end of the Introduction. Conversely, for the above group, these three criteria are made worse when canceling the vibration. Therefore, the above group allows to test the first and third hypotheses presented at the end of the Introduction.”

Comment 3: Fig 1C: Does the dashed line represent the final posture of the exoskeleton for different trial? Or it just connect the start and end position of the exoskeleton? I do have trouble understanding the plots for a long time and I think it could be make more clear for easier understanding.

We have now clarified that these are illustrative successive trajectories, represented as straight lines for the sake of simplicity, in the caption of Fig. 1C.

References

- [1] A. Garg, K. Hegmann, and J. Kapellusch, “Short-cycle overhead work and shoulder girdle muscle fatigue,” International Journal of Industrial Ergonomics, vol. 36, no. 6, pp. 581–597, Jun. 2006.
- [2] P. Maurice, V. Padois, Y. Measson, and P. Bidaud, “Experimental assessment of the quality of ergonomic indicators for dynamic systems computed using a digital human model,” International Journal of Human Factors Modelling and Simulation, vol. 5, no. 3, p. 190, 2016.
- [3] T. J. Armstrong, P. Buckle, L. J. Fine, M. Hagberg, B. Jonsson, A. Kilbom, I. A. Kuorinka, B. A. Silverstein, G. Sjøgaard, and E. R. Viikari-Juntura, “A conceptual model for work-related neck and upper-limb musculoskeletal disorders.” Scandinavian Journal of Work, Environment & Health, vol. 19, no. 2, pp. 73–84, Apr. 1993.

- [4] C. R. Dickerson, A. C. McDonald, and J. N. Chopp-Hurley, "Between two rocks and in a hard place: reflecting on the biomechanical basis of shoulder occupational musculoskeletal disorders," Human Factors: The Journal of the Human Factors and Ergonomics Society, vol. 65, no. 5, pp. 879–890, Jan. 2020.
- [5] L. McAtamney and E. N. Corlett, "RULA: a survey method for the investigation of work-related upper limb disorders," Applied Ergonomics, vol. 24, no. 2, pp. 91–99, apr 1993.
- [6] S. Hignett and L. McAtamney, "Rapid entire body assessment (REBA)," Applied Ergonomics, vol. 31, no. 2, pp. 201–205, apr 2000.
- [7] B. Hwang and D. Jeon, "A method to accurately estimate the muscular torques of human wearing exoskeletons by torque sensors," Sensors, vol. 15, no. 4, pp. 8337–8357, apr 2015.
- [8] D. Verdel, S. Bastide, N. Vignais, O. Bruneau, and B. Berret, "Human weight compensation with a backdrivable upper-limb exoskeleton: identification and control," Frontiers in Bioengineering and Biotechnology, vol. 9, pp. 1–16, jan 2022.
- [9] E. Todorov and M. Jordan, "A minimal intervention principle for coordinated movement," Advances in neural information processing systems, vol. 15, 2002.

Answer to reviewers:

We would like to thank the reviewers and editors for their constructive comments that contributed to greatly improve the manuscript. We provide below a point-to-point response to the reviewers' comments. Our answers appear in blue and the modifications in the revised manuscript are also highlighted in blue for the sake of clarity.

1 Reviewers #1 and #2

General assessment: The authors have clearly made an effort to improve the manuscript in response to earlier feedback. The revised version is more transparent and easier to follow, and it reflects meaningful progress in clarifying the contributions. That said, there remain some issues that need to be addressed before the manuscript can be considered for publication. These issues concern both methodological clarity and writing quality, which are required to make the manuscript suitable for publication.

We thank the reviewers for their thorough assessment of our work. We have included a point-by-point answer to their comments below.

Statistical analysis: Our biggest concern at this stage is that there is no explicit statement in the statistical analysis section confirming that the assumptions of linear mixed models were verified. Specifically, the authors should make a statement if they checked:

- The explanatory variables are linearly related to the response,
- The errors have constant variance (homoscedasticity),
- The errors are independent, and
- The errors are normally distributed.

Without this information, the validity of the statistical conclusions remains uncertain.

We agree with the reviewers that a statement was needed in the statistics section. Therefore, we now added the following:

“Note that we verified the linearity, homoscedasticity, normality and independence of the errors in our distributions to ensure the validity of the tests.”

We provide a few examples of the verifications that we conducted for all the parameters in the Figure below.

Figure 1: Illustration of conducted verifications of LMM hypotheses for the gravity related parameter. Verifications are illustrated for δ_G (top) and the normalized gravity-related torques (bottom), which are two of the main parameters of the study. **A,D.** Normality hypothesis: residuals are close to the expected 45° line, as expected for a normal distribution. **B,E.** Linearity and homoscedasticity hypotheses: residuals are evenly spread around 0, confirming these hypotheses. **C,E.** Independence hypothesis: residuals do not exhibit any specific pattern across the different conditions, confirming their independence.

Readability: In addition, we believe a general revision of the writing is still necessary. The manuscript remains dense in places, and some sentences need to be re-read to be fully understood. Connectors are sometimes overused, which makes the text unnecessarily heavy. For example, the word “importantly” appears multiple times in close succession. A careful stylistic revision would improve the manuscript’s flow.

We agree with the reviewers. We have performed a thorough check to simplify the sentences and make the whole text more readable. In the marked version, all the changes are highlighted in blue with the previous text barred to simplify the assessment. Note that due to the barred texts, the positioning of the figures is suboptimal in the marked version.

Research questions (Lines 82–83): In my view, RQ2 and RQ3 could be merged into a single, clearer question: “Do participants retain learning from the exoskeleton?” The two-group design is not a separate research question but rather a methodological approach to address this issue, clarifying whether the observed effects result from adopting a naturally less effortful posture or from actual retention of learning under a less natural condition. Presenting them as distinct RQs therefore adds unnecessary complexity without improving clarity.

We thank the reviewers for this elegant suggestion. The new hypotheses read as (p.2, l.81-86):

*“The investigations focus on verifying ~~three~~two hypotheses: (i) whether participants succeed or fail to adapt by adopting a posture minimizing the intensity of vibrations and/or of gravity-related efforts, (ii) whether they can **learn and** retain ~~the~~adapted postures ~~when it is better than~~different from their nominally preferred one, which would be critical for ergonomics improvement (see Fig. 1C), ~~and~~ (iii) ~~whether the adaptation to the vibrations applied by the exoskeleton is robust, i.e. can also happen when the targeted posture is more effortful than the participant’s preferred posture.~~”*

Minor comments: Figure 1C caption: The authors refer to “top” and “bottom,” but the figure is displayed horizontally. Using “left” and “right” would be clearer and avoid confusion. Ergonomic framing: Although the addition of ergonomics was requested, the authors now repeat the ergonomic assumptions three times, which makes the text repetitive for the reader.

We have corrected the caption of the figure (see p.3). Furthermore, we have removed the details on ergonomic framing from the results, and the detailed description from the Discussion, so that it is only developed in the Introduction.

2 Reviewer #3

I think the revised manuscript adequately addressed all my previous questions and comments. I have no additional comments.

We thank the reviewer for their assessment of our manuscript.